



# Multi-objective calibration and evaluation of the ORCHIDEE land surface model over France at high resolution

Peng Huang[1], Agnès Ducharne[1], Lucia Rinchiuso[1], Jan Polcher[2], Laure Baratgin[2], Vladislav Bastrikov[3], and Eric Sauquet[4]

[1]Sorbonne Université/CNRS/EPHE, METIS-IPSL, Paris, France
[2]CNRS/Sorbonne Université/École Polytechnique, LMD-IPSL, Palaiseau, France
[3]Science Partners, Paris, France
[4]INRAE, UR RiverLy, Villeurbanne, France

**Correspondence:** Peng Huang (peng.huang@sorbonne-universite.fr)

**Abstract.** We present here a strategy to obtain a realistic hydrological simulation over France with the ORCHIDEE land surface model. The model is forced by the Safran atmospheric reanalysis at 8-km resolution and hourly time steps from 1959 to 2020, and by a high-resolution DEM (around 1.3 km in France). Each Safran grid cell is decomposed into a graph of hydrological transfer units (HTUs) based on the higher resolution DEM to better describe lateral water movements. In particular, it is possible to accurately locate 3507 stations among the 4081 stations collected from the national hydrometric network HydroPortail (filtered to drain an upstream area larger than 64 km$^2$). A simple trial-and-error calibration is conducted by modifying selected parameters of ORCHIDEE to reduce the biases of the simulated water budget compared to the evapotranspiration products (the GLEAM and FLUXCOM datasets) and the HydroPortail observations of river discharge. The simulation that is eventually preferred is extensively assessed with classic goodness-of-fit indicators complemented by trend analysis at 1785 stations (filtered to have records for at least 8 entire years) across France. For example, the median bias of evapotranspiration is -0.5% against GLEAM (-4.3% against FLUXCOM), the median bias of river discharge is 6.3%, and the median KGE of square-rooted river discharge is 0.59. The spatial contrasts and temporal trends of river discharge across France are well represented with an accuracy of 76.4% for the trend signal and an accuracy of 62.7% for the trend significance. Despite inadequate performance in some specific regions (the Alps and the Seine sedimentary basin), this study offers a thorough historical overview of water resources and a robust configuration for climate change impact analysis at the nationwide scale of France.

## 1 Introduction

Land surface models (LSMs) are the land surface components of Earth system models (ESMs) and simulate water, energy and carbon fluxes across continents. The offline use of LSMs as independent physically-based distributed hydrological models has emerged in the recent decades for evaluating water resources and investigating climate change impacts at both regional and global scales (e.g., Cai et al., 2014; Pokhrel et al., 2021; Telteu et al., 2021).

Since LSMs provide the lower boundary to atmosphere circulation in ESMs, the spatial discretization of LSMs is usually consistent with the atmospheric component or atmospheric forcing (in offline mode) so that spatial discontinuity can be averted



during land-atmosphere coupling. However, the spatial resolution of the atmosphere component, typically 0.5° (IPCC, 2023), or the atmospheric forcing, typically tens of kilometres (e.g., EuroCORDEX at 11 km in Jacob et al. 2014 and ERA-5 at 31

km in Hersbach et al. 2020), are too coarse to represent topographic details. In addition, river flows are much more constrained by topographic conditions than by the atmosphere. Thus, accurate hydrological simulations for flood risk assessment, drought monitoring and human impact assessment (e.g., dam regulation and irrigation) are difficult to achieve, especially for local-scale implementations. Hence, incorporating high-resolution river routing systems in LSMs is necessary to improve hydrological simulations by better characterizing the morphological conditions of river basins with high-resolution digital elevation models

(DEMs) (Bierkens et al., 2015).

The conventional approach to computing river basin discharges relies on independent runoff routing models (RRMs) that interpolate the lateral water flows simulated by the LSM to the grid cells of the RRMs and cascade river discharges along the drainage network. The RRMs represent the horizontal movements of water fluxes while their vertical movements are kept in LSMs. There are many different RRMs in the literature, such as the TRIP at 0.5° resolution (Oki and Sud, 1998; Oki et al.,

1999; Vergnes and Decharme, 2012), the RiTHM at 0.25° resolution (Ducharne et al., 2003), and the HYDRA at 5' resolution (Coe, 2000). To bridge the gap between high-resolution hydrological simulations and coarse LSM grid cells, the concept of constructing hydrological transfer units (HTUs) in LSM grid cells with high-resolution DEMs was proposed to better represent natural river systems (Nguyen-Quang et al., 2018; Polcher et al., 2023). These hydrologically consistent units within each atmospheric grid cell are connected to horizontally transfer the simulated lateral flows so that the generated river flows in

one atmospheric grid cell can flow into neighbouring grid cells (HTU to HTU and then grid cell to grid cell) (Polcher et al., 2023). The vertical and horizontal movements of water fluxes can be maintained within the LSM instead of separating these two movements by two models (i.e., LSMs and RRMs) at different resolutions, which facilitates the representation of human impacts on hydrological processes (Zhou et al., 2021; Baratgin et al., 2024).

When using LSMs to simulate realistic water fluxes, including river discharge, a calibration step is often necessary: even if

LSMs are designed to be as physical as possible, they inevitably contain parameters that are hard to measure directly, such as vegetation water stress (Ruiz-Vásquez et al., 2023), or soil properties at different depths (Yang et al., 2016). Traditional hydrologic calibration is primarily conducted against river discharge observations (e.g., Troy et al., 2008; Gou et al., 2020; Cho and Kim, 2022; Rummler et al., 2022), which is considered to be a well-suited benchmark (Prentice et al., 2015). However, calibrating physically-based LSMs to discharge alone does not guarantee the realistic representation of hydrological processes and the accurate simulation of other LSM outputs, such as soil moisture (Sutanudjaja et al., 2014) and evapotranspiration (Rajib et al.,

accurate simulation of other LSM outputs, such as soil moisture (Sutanudjaja et al., 2014) and evapotranspiration (Rajib et al., 2018a, b). Over the recent decades, the advancement of reanalysis and remote-sensing data quality at fine scales, such as snow cover (Hall et al., 2002), evapotranspiration (Martens et al., 2017) and soil moisture (Dorigo et al., 2017) products, provides new opportunities to investigate and improve the effectiveness of LSMs in representing water fluxes. Multi-objective calibration works that combine discharge observations with these data products have shown an overall improvement on hydrological simulation performance (e.g., López López et al., 2017; Jiang et al., 2020; Yang et al., 2021).

simulation performance (e.g., López López et al., 2017; Jiang et al., 2020; Yang et al., 2021).

The complexity of LSMs makes their calibration extremely difficult in practice because their large number of parameters induces high degrees of freedom (Fisher and Koven, 2020). Two kinds of methods are mostly used to adjust the parameter set



of LSMs: automatic (i.e., optimization techniques) and manual (i.e., trial-and-error procedure). Numerous efforts have been put into optimization algorithms (e.g., Müller et al., 2015; Yang et al., 2021; Cheng et al., 2023), with the major challenge being the

computational burden, especially for high-resolution applications (Bierkens et al., 2015; Sun et al., 2020). Another limitation of the automatic calibration method, especially if applied to large parameter sets, stems from the equifinality issue that many different parameter sets lead to equally good results (Beven, 2006; Fisher and Koven, 2020). In manual calibration practice, modelers select a few parameter sets, apply them to run the LSM, and choose the best parameter set based on evaluations. Albeit highly dependant on expert judgement, this method can be efficient in saving model run time compared to the hundreds

of model runs required by automatic methods (Schaperow et al., 2021). Either way, a perfect calibration is always impossible to achieve due to inherent uncertainties in forcing data (e.g., Gelati et al., 2018; Kabir et al., 2022) and benchmark observations (e.g., Zeng and Cai, 2018).

Hydrological model performance is typically evaluated with goodness-of-fit indicators, such as Kling-Gupta (Kling et al., 2012) or Nash-Sutcliffe (Nash and Sutcliffe, 1970) efficiencies. In doing so, discharge is often transformed for performance

evaluation, such as with logarithm to emphasize on low flows (Santos et al., 2018) or with square-root to balance low/high flows (Song et al., 2019). Hydrological signatures that characterize statistical or dynamic features of discharge (e.g., annual discharge and low flow duration) can also be used to evaluate simulation performance (see the review by McMillan (2021)). These traditional indicators implicitly assume stationary conditions and are no longer sufficient since "stationarity is dead" (Milly et al., 2008). As shown by Todorović et al. (2022), the traditional indicators do not guarantee the reproduction of

streamflow trends with hydrological models. Thus, trend analyses are important to evaluate the robustness of hydrological models over the long term, which is essential to subsequent applications for climate change assessment (Fowler et al., 2020).

At the nationwide scale of France, the first distributed LSM for hydrological applications has been proposed by Habets et al. (2008). It couples the Safran atmospheric reanalysis system (Vidal et al., 2010a) and the ISBA LSM (Decharme and Douville, 2006), both at a spatial resolution of 8 km x 8 km, and the MODCOU model (Ledoux et al., 1989) for groundwater and river

flow, with a variable resolution down to 1 km along rivers. This model, called SIM for Safran-ISBA-MODCOU, was validated by comparison to hundreds of hydrometric stations with a focus on the Seine, Loire, Garonne, and Rhône River basins, the four major river basins in France from 1995 to 2005. Observations of piezometric head and snow depth at several sites are also used to evaluate the SIM model. Since then, it was used to assess drought risks in the atmosphere, soils and rivers, and to investigate the impact of climate change for future adaptation at the national scale (e.g., Quintana Seguí et al., 2009; Vidal

et al., 2010b; Boé et al., 2009), and recently improved to better describe the water and energy budgets in France from 1958 to 2018, providing an extensive historical analysis (Le Moigne et al., 2020).

The main goal of the present study is to obtain a realistic and robust hydrological simulation over France with another LSM, the Organising Carbon and Hydrology in Dynamic Ecosystems (ORCHIDEE) LSM, using a high-resolution HTU-based routing scheme (Nguyen-Quang et al., 2018; Polcher et al., 2023). This approach allowed us to simulate river discharge at thou-

sands of hydrometric stations across French rivers, supporting a thorough performance evaluation. In doing so, we combined traditional indicators implicitly assuming stationary conditions, and indicators about trend accuracy because "stationarity is dead" (Milly et al., 2008). Another novelty stems from a multi-objective calibration focused on the water budget, benefiting





from river discharge observations at 1785 hydrometric stations, and evapotranspiration products. The parameterization selected in this study has been used to project the impact of climate change on French water resources, which contributes to the national

Explore2 project (https://professionnels.ofb.fr/fr/node/1244) for future adaptation design. Here, we only present the simulation results from 1959 to 2020 to assess the performance of the ORCHIDEE LSM. In section 2, the ORCHIDEE LSM is presented, followed by a summary of the input data, the benchmark datasets, and the calibration strategy. In section 3, we detail the calibration procedure that was conducted step by step to improve the overall simulated water budget and evaluate the simulation which was eventually chosen with classic goodness-of-fit measures and trend analysis; finally, a discussion and conclusions

are presented.

## 2 Materials

### 2.1 The ORCHIDEE LSM (revision 7738)

The ORCHIDEE model is a physically-based LSM developed at the Institut Pierre Simon Laplace (IPSL). It is the land component of the IPSL climate model (Boucher et al., 2020; Cheruy et al., 2020), which is used for all the past and future climate

simulation exercises carried out for the IPCC reports as part of the Coupled Model Intercomparison Project (CMIP). In this study, the ORCHIDEE model is not coupled to the IPSL climate model (off-line simulation) but is instead fed by an atmospheric forcing (section 2.2.1). The ORCHIDEE model couples the SECHIBA (water and energy budgets in Ducoudré et al., 1993) and STOMATE (carbon budget and phenology in Krinner et al., 2005) modules. This coupling describes the hydrological processes (e.g., soil moisture diffusion, evapotranspiration, and river discharge) and their interactions with vegetation and

the carbon cycle so that the simulated variables depend on the atmospheric $CO_2$ concentration. The water, energy and carbon fluxes are calculated on a 30-minute time step within each atmospheric grid cell, and the river discharges are then deduced by aggregating the lateral flows of each grid cell along the river routing network.

The vegetation in a grid cell is not uniform but rather comprises a mosaic of several plant function types (PFTs, section 2.2.2). Table A1 summarizes the 15 PFTs used in the ORCHIDEE LSM. Each PFT is characterized by different morphological,

physiological, phenological and radiative properties, mainly based on specialized literature. The root density profile of each PFT in the ORCHIDEE model is assumed to decrease exponentially with depth and is modulated by a decay factor $c$, as shown in Figure F1. The root density of each PFT can be increased (decreased) by decreasing (increasing) $c$. Crop and grass PFTs have higher $c$ values than forest PFTs; the roots of crop and grass PFTs are concentrated in surface soil layers while the roots of forest PFTs can pass through deep soil layers.

The soil is 2 m deep, and each grid cell is characterized by the dominant soil texture (section 2.2.2). The soil water retention properties (including porosity $\theta_s$, field capacity $\theta_c$ and wilting point $\theta_w$) depend on the texture as detailed in Table B1. The soil hydraulic conductivity at saturation $K_s$ is not vertically constant, as shown in Figure D1: ORCHIDEE assumes an exponential decrease with depth due to soil compaction, ruled by a decay factor $f$, combined with an increase towards the soil surface due to the presence of roots, which enhances infiltration capacity (de Rosnay et al., 2002; d'Orgeval et al., 2008). This effect depends

on the root density profile. At each time step, soil moisture is redistributed vertically according to the Richards equation (flow in



an unsaturated medium) and discretized into 22 layers over 2 m, taking into account surface boundary conditions by infiltration and soil evaporation, withdrawals by roots through the entire soil depth to supply transpiration, and gravitational drainage at the bottom of the soil (Tafasca et al., 2020). In this framework, transpiration depends on soil moisture and a factor $p$, which represents the soil moisture content above which transpiration is maximal, i.e., not limited by water stress. Figure E1 shows
how the parameter $p$ constrains transpiration.

    Evapotranspiration (ET) is calculated as the sum of plant transpiration, evaporation of intercepted water, soil evaporation and snow sublimation. This calculation does not depend on potential evapotranspiration, but it is coupled to the surface energy balance, which requires a sub-hourly time step (here, 30 minutes) to describe the diurnal radiation cycle. The four ET fluxes in ORCHIDEE are described by a bulk aerodynamic formulation, in which the roughness length for momentum $z_{0m}$ and for
heat $z_{0h}$ control the aerodynamic resistance. $z_{0m}$ and $z_{0h}$ in ORCHIDEE can be calculated by prescribed parameters: $z_{0m}$ is calculated by a first-order approximation of vegetation height, i.e., multiplied by a factor $f_z$ (e.g., 1/10 in Brutsaert, 2005); $\frac{z_{0m}}{z_{0h}}$ is parameterized as 1 in CMIP5 to compensate for forcing errors (Dufresne et al., 2013). Note that $\frac{z_{0m}}{z_{0h}}$ should be larger than 1; for example, $\frac{z_{0m}}{z_{0h}}$=10 was proposed by Brutsaert (2005). $z_{0m}$ and $z_{0h}$ in the ORCHIDEE can also be calculated by the dynamic (dyn) method proposed by Su et al. (2001) as implemented in CMIP6 (Boucher et al., 2020), and the formulation of
the method as well as its application is detailed in Su et al. (2001) and Su (2002). This dynamic method generally decreases ET simulation for the CMIP6 configuration of ORCHIDEE compared with the CMIP5 configuration.

    Snowpack and its dynamics are described by a 3-layer model, which makes it possible to account for variations in albedo, density and thus the insulating properties of the snowpack as a function of the age of the snow and the nature of the underlying vegetation (Wang et al., 2013). Rainfall not intercepted by vegetation cover and meltwater can infiltrate or runoff when the
rainfall exceeds the surface hydraulic conductivity. The two runoff terms, surface runoff and gravitational drainage at the bottom of the soil, are summed as the total runoff.

    Eventually, river flows are calculated by a high-resolution routing model (Nguyen-Quang et al., 2018; Polcher et al., 2023), which aggregates the total runoff of HTUs within each atmospheric grid cell along the river routing network (section 2.2.3). The routing model contains 3 linear reservoirs in each HTU: the river reservoir allows horizontal flow to transfer from HTU
to HTU along the high-resolution river network; the groundwater and surface water reservoirs are used to compute the mean transit time of groundwater drainage and surface runoff, respectively, and their contribution to the outflow from the river reservoir (Schrapffer et al., 2020). The groundwater reservoir is simplified as a free aquifer, and thus the outflow is the base flow. The resident time of each reservoir depends on the length and slope of the HTU, modulated by a time constant specific for each reservoir (Polcher et al., 2023), which leads to a duration of resident time from long to short according to the order of
groundwater, surface water, and river reservoirs. In this context, the flow velocity at each HTU in the river reservoir does not vary with discharge and does not depend on flooding, which is not explicitly described.



## 2.2 Input data over France

### 2.2.1 Atmospheric forcing

The near-surface meteorological Safran reanalysis with a spatial resolution of 8 km and a temporal resolution of hourly time steps (Vidal et al., 2010a) is used in this study to drive the ORCHIDEE simulations over France. The Safran grid cell is thus the horizontal resolution of the ORCHIDEE simulations. To cover the complete drainage area of French rivers, especially in the Eastern alpine parts of France, the Safran reanalysis was extended to some parts of neighbouring countries, especially Switzerland (Figure 1). The Safran reanalysis is available from 08/1958 onwards and contains atmospheric data for ORCHIDEE: air temperature, air pressure, air specific humidity, wind speed, liquid and solid precipitation, and downwards longwave and shortwave radiation. In addition, annual $CO_2$ concentration observations in the atmosphere are sourced from Lurton et al. (2020).

### 2.2.2 Boundary conditions: soil, vegetation and land use

The annual vegetation and land use maps for France used in this study are sourced from Lurton et al. (2020) and derived from two products (see https://orchidas.lsce.ipsl.fr/dev/lccci/ for further data aggregation information): the global ESA CCI vegetation distribution (Harper et al., 2023) at 300 m spatial resolution is used to generate 15 generic PFTs gridded at 0.25° spatial resolution for the ORCHIDEE model (bare soil, 2 crop PFTs, 4 grass PFTs, and 8 forest PFTs as detailed in Table A1) to represent land cover distribution; the generic 15 PFT distribution is then combined with the Land-Use Harmonization 2 (LUH2) dataset (Hurtt et al., 2020) at 0.25° spatial resolution to produce the temporal evolution of 15 PFTs for CMIP6 simulations over the historical and future periods (850-2100). This information is ultimately reaggregated into Safran grid cells to account for the spatial distribution of the PFTs across France (Figure A1). The harvested wood biomass is also sourced from the LUH2 dataset.

The sensitivity of the ORCHIDEE model to global and regional soil texture maps was tested in the literature (e.g., Tafasca et al., 2020; Kilic et al., 2023). The global soil texture map of Reynolds et al. (2000) shows a better hydrological performance for the Seine River basin (Kilic et al., 2023). Therefore, the soil texture data for France used in this study are based on the global soil texture map of Reynolds et al. (2000) at a 1/12° spatial resolution, which classifies soil textures into 12 USDA types. The soil texture information over France is then rescaled into Safran grid cells by keeping the dominant soil texture type for each grid cell (Figure B1). The dominant soil textures in France are loam and clay loam according to the Reynolds soil texture map.

### 2.2.3 River routing network

The construction of the river routing network across France for the ORCHIDEE model is based on the DEM at 1/60° resolution (approximately 1.3 km over France) built by upscaling the MERIT Hydro global hydrography map at 3-arc sec resolution (Yamazaki et al., 2019) with the Iterative Hydrography Upscaling (IHU) method (Eilander et al., 2021). The DEM incorporates the following topographic and hydrologic information from the MERIT Hydro dataset: elevation, flow direction, flow accu-



**Figure 1.** The topography and hydrography of Metropolitan France (including Corsica), delineated in black contours, extended to neighbouring countries based on the upscaled MERIT hydro DEM. The river networks in blue are represented by the pixels with a flow accumulation larger than 200 pixels. Four major river basins, the Seine, the Loire, the Rhône and the Garonne, and three major mountain ranges, the Alps, the Massif central and the Pyrenees, in France are marked on the map. The gray points represent the central points of the Safran reanalysis grid cells.





mulation and distance to the ocean for each pixel, as shown in Figure 1; this information is used to construct the river routing network connected by the HTUs of Safran grid cells as detailed in Polcher et al. (2023). In this study, we selected 15 trunca-

tions to construct HTUs (nbmax = 15, i.e., the maximum number of HTUs within each atmospheric grid cell) given the spatial resolutions of the DEM and Safran. Within this framework, the hydrometric stations collected from the Explore2 project are positioned on the constructed high-resolution river routing network to comply with the following criteria: the distance between the real station and the modelled station must be less than 5 km and the error of the upstream surface at the modelled station must be less than 20%. Of the 4081 hydrometric stations collected, 3507 stations are within the above tolerance (86% of the

total stations). Finally, ORCHIDEE can monitor the flow out of the HTU associated with the station during the simulation.

## 2.3   Evaluation strategy

Three evaluation datasets are used in this study to assess the performance of the ORCHIDEE simulation. The GLEAM dataset (Martens et al., 2017) provides daily ET data at a 0.25° resolution at the global scale from 1980 to 2020; these data were derived from the Priestley and Taylor (1972) evaporation model with satellite-based products (net radiation, precipitation, surface

soil moisture, skin and air temperatures, vegetation optical depth, and snow water equivalent) as inputs. The FLUXCOM dataset (Jung et al., 2019) provides daily ET at a 0.5° resolution from 2001 to 2015 based on machine learning algorithms that merge global FLUXNET measurements with remote sensing and meteorological observations. Both ET benchmarks are aggregated to Safran grid cells at monthly time steps. The Explore2 project provides records of daily river discharge (Q) across France extracted from the French national hydrometric HydroPortail (Leleu et al., 2014; Delaigue et al., 2020). Of the 3507

hydrometric stations placed across the constructed river routing network, 1785 stations had Q records for at least 8 entire years over the simulation period of 1959-2020; these records were used to calibrate and evaluate the ORCHIDEE simulation.

   The criteria bias, Pearson correlation, Kling-Gupta efficiency (KGE, Kling et al., 2012) and timelag are used to quantitatively evaluate the goodness-of-fit of the simulated ET and Q against the above datasets. The timelag calculation is based on cross-correlation and determined by a value t corresponding to the maximum correlation (a positive t means that the simulated Q time series lags the observed time series by t days; a negative t means that the simulated Q time series leads the observed time

series by -t days). In addition, to verify the ability of the ORCHIDEE LSM to reproduce the observed trends of ET and Q, the trend signals of the 1785 selected stations were calculated using the Sen's slope estimator (Sen, 1968), and the nonparametric Mann-Kendall test (Mann, 1945; Kendall, 1948) was used to determine the significance of the calculated trend signals. The significance level in this study is set to 5%.

## 2.4   Calibration design

The simulation period is from 1959 to 2020, with a warm-up from 1959 to 1968 to provide reasonable initial conditions, and the output variables (e.g., ET and Q) are aggregated to daily time steps. The calibration design is based on iterative trial-and-error procedures of the ORCHIDEE parameters to minimize the biases of ET and Q, and 6 calibration experiments are illustrated in Table 1 to show the gradual decrease of both biases. STD is the parameter set sourced from CMIP6 (Boucher et al., 2020) as the

start of the calibration design, and STD calculates ET via a dynamic method. STD forced with Safran reanalysis significantly



underestimates ET and overestimates Q; the following calibration experiments focus on changing the parameters that increase ET and decrease Q. EXP1a and EXP1b calculate ET by the prescribed $\frac{z_{0m}}{z_{0h}}$ and $f_z$, respectively, and the values are suggested by Brutsaert (2005). EXP2 increases the decay factor $f$, and the soil hydraulic conductivity decreases more rapidly with depth so that the soil drainage decreases (Q decreases) and ET increases. EXP3 decreases the soil water threshold for transpiration from 0.8 to 0.5, and transpiration (thus, ET) increases. EXP4 changes the root profiles of 15 PFTs by increasing the $c$ parameter of tree and boreal grass PFTs to decrease their root density and decreasing the $c$ parameter of crop PFTs to increase crop root density. As such, the transpiration of trees and boreal grasses decreases, while the transpiration of crops increases.

**Table 1.** Parameter sets applied to the calibration experiments of the ORCHIDEE LSM. The means of daily evapotranspiration (ET), surface runoff ($R_s$), drainage ($R_d$) and total runoff (R) are calculated over the extended Safran coverage and the 1959-2020 simulation period. The median of ET bias is calculated for all Safran grid cells of the extended Safran coverage with FLUXCOM dataset over the 2001-2015 period. The median of river discharge (Q) bias is calculated with the 1785 selected French hydrometric stations over the 1959-2020 period.

| | | Roughness | | Hydraulics | Vegetation | | ET | $R_s$ | $R_d$ | R | Bias ET | Bias Q |
|---|---|---|---|---|---|---|---|---|---|---|---|---|
| | | $\frac{z_{0m}}{z_{0h}}$ | $f_z$ | $f$ | $p$ | $c$ | [mm/d] | [mm/d] | [mm/d] | [mm/d] | [%] | [%] |
| | STD | dyn | 1/15 | 2 | 0.8 | ref | 1.350 | 0.330 | 0.933 | 1.263 | -14.9 | 28.4 |
| Calibrations | EXP1a | 10 | - | - | - | - | 1.453 | 0.314 | 0.847 | 1.161 | -9.2 | 16.7 |
| | EXP1b | 10 | 1/10 | - | - | - | 1.471 | 0.318 | 0.824 | 1.143 | -8.0 | 13.6 |
| | EXP2 | 10 | 1/10 | 4 | - | - | 1.490 | 0.916 | 0.208 | 1.124 | -6.8 | 11.7 |
| | EXP3 | 10 | 1/10 | 4 | 0.5 | - | 1.498 | 0.910 | 0.206 | 1.116 | -6.2 | 10.7 |
| | EXP4 | 10 | 1/10 | 4 | 0.5 | new | 1.526 | 0.893 | 0.195 | 1.089 | -4.3 | 6.3 |

- same as STD; dyn as dynamic;

ref = [5.0, 0.8, 0.8, 1.0, 0.8, 0.8, 1.0, 1.0, 0.8, 4.0, 4.0, 4.0, 4.0, 4.0, 4.0] in Table A1;

new = [5.0, 0.8, 0.8, 1.0, 0.8, 1.5, 2.0, 2.0, 1.5, 4.0, 4.0, 2.0, 2.0, 4.0, 6.0] in Table A1.

# 3 Results

## 3.1 Evaluation of simulated basin area

Figure 2 shows the good performance of the high-resolution river routing model in simulating the basin areas across France, with $R^2$ = 0.999 across the 3507 stations. Classically, the performance increases with increasing river basin area, with $R^2$ ranging from 0.992 for basins less than $10^3$ km$^2$ to 0.998 for basins larger than $10^4$ km$^2$. However, the routing model tends to overestimate basin areas for basins less than $10^4$ km$^2$ but tends to slightly underestimate basin areas for basins larger than $10^4$ km$^2$. There is no significant positive or negative bias in the simulated basin area for the 4 major river basins (the Seine, the Loire, the Rhône, and the Garonne), and the biases of most of the simulated basins are less than 5%. For basins larger






**Figure 2.** The comparison between the simulated upstream basin area and the reference area in HydroPortail for the 3507 stations located in the high-resolution river networks: (a) scatter plot of simulated area to reference area, (b) boxplot of simulated area bias and (c) spatial map of simulated area bias for basins less than $10^3$ km$^2$, between $10^3$ km$^2$ and $10^4$ km$^2$, and larger than $10^4$ km$^2$.





than $10^3$ km$^2$, most of the biases are larger than 5% are located in the mountainous regions, especially in the Alps, given the complicated topography.

## 3.2 Calibration results

Figure 3 illustrates how the performance criteria of the simulated ET and Q improve during the calibration experiments.

The first three calibration experiments show the impact of two different methods in calculating ET. STD applies a dynamic physically-based model that calculates $z_{0m}$ and $z_{0h}$ with the variables simulated by ORCHIDEE (e.g., canopy height, LAI, and fractional coverage for 15 PFTs). EXP1a and EXP1b prescribe $\frac{z_{0m}}{z_{0h}}$ and $f_z$ values to approximate $z_{0m}$ and $z_{0h}$ only with the simulated variable of canopy height. Compared with STD, EXP1a decreases the negative ET bias against FLUXCOM by 5.7% by greatly increasing $z_{0h}$ from $1.10 \times 10^{-4}$ to $1.43 \times 10^{-2}$ m over the extended simulation domain and simulation period

($z_{0m}$ decreases from 0.385 to 0.255 m). Compared with EXP1a, EXP1b decreases the negative ET bias by 1.2% by increasing $f_z$ from 1/15 to 1/10: $z_{0m}$ increases from 0.255 to 0.482 m, and $z_{0h}$ increases from $1.43 \times 10^{-2}$ to $2.28 \times 10^{-2}$ m. Figures C1 and C2 show the spatial and temporal changes of the $z_{0h}$, $z_{0m}$ and ET values for the first three calibration experiments. Q is thus decreased due to the water budget of ORCHIDEE when ET is increased, and the positive Q bias against HydroPortail decreases by 11.7% and 3.1%, respectively. For the first three calibration experiments, the biases of simulated ET and Q against

observation datasets are gradually decreased and EXP1a decreases the biases of the simulated ET and Q the most. In addition, the KGE values of square-rooted Q against observations are slightly increased. However, the correlation values of the simulated Q against observations are slightly decreased, and the simulated Q tends to gradually lag behind the observations for the first three calibrations.

To improve the goodness-of-fit of the simulated Q, EXP2 increases the decay factor $f$ from 2 to 4 compared to EXP1b,

which decreases the hydraulic conductivity for soil layers below 0.3 m, while the hydraulic conductivity for soil layers above 0.3 m remains unchanged (Figure D2a-b). A decrease of hydraulic conductivity in deep soil layers leads to a decrease of drainage at the soil bottom. Since less water drains from deep soil layers while the surface soil layers maintain the same infiltration capacity, the latter can saturate more easily, and more surface runoff is produced (Table 1). Eventually, the total runoff is decreased from EXP1b to EXP2 as the surface runoff increase is smaller than the decrease of gravitational drainage

at the soil bottom. EXP2 thus decreases the positive Q bias against HydroPortail (by 1.9%), as well as the negative ET bias against FLUXCOM (by 1.2%). In addition, the ratio of surface runoff to total runoff is greatly increased from 27.9% in EXP1b to 81.5% in EXP2, which results in more "fast" surface flow and less "slow" groundwater, leading to more responsive Q to precipitation events. The correlation and KGE criteria of the simulated Q are improved from 0.59 to 0.69 and from 0.54 to 0.59, respectively (Figure 3). The timelag criterion of the simulated Q is also greatly improved from a range of -11 to 27 days

to a range of -3 to 5 days. Similar improvements in streamflow dynamics could be obtained by changing the time constant of the fast and slow routing reservoirs without improving the Q and ET biases.

Two additional simulations, EXP3 and EXP4, were conducted to further improve the bias criteria by changing the vegetation parameters in ORCHIDEE to potentially increase transpiration (thus ET): EXP3 reduced the soil moisture stress for transpi-





**Figure 3.** Performance criteria of calibration experiments: (a) bias of the simulated ET to GLEAM dataset; (b) bias of the simulated ET to FLUXCOM dataset; (c) bias of the simulated Q; (d) Pearson correlation coefficient of the simulated Q; (e) KGE of the square-rooted simulated Q; (f) timelag of the simulated Q. The calculation of biases for the simulated ET is applied to all Safran grid cells of the extended domain over the 1980-2020 period against GLEAM dataset and over the 2001-2015 period against FLUXCOM dataset, respectively. The calculation of criteria for the simulated Q is applied to the 1785 hydrometric stations in the HydroPortail dataset with records for at least 8 entire years.





ration, while EXP4 changed the vegetation root profile. Transpiration is conveyed by the factor Us in ORCHIDEE, which is
negatively related to the water stress factor F and positively related to root density.

The factor F depends on soil moisture and on a threshold parameter $p$, as illustrated in Figure E1: there is no soil moisture
stress if F = 1, which occurs when the soil moisture exceeds $\theta_w + p \times (\theta_c - \theta_w)$. By decreasing $p$ from 0.8 in EXP2 to 0.5 in
EXP3, a wider range of soil moisture leads to F = 1 and thus unstressed transpiration. As shown in Figure E2, transpiration
(Us) is increased for all PFTs and the effect is more pronounced for crop PFTs (PFTs 12 and 13) than for forest PFTs (PFTs 7
and 8). However, this general decrease of soil water stress to favour transpiration is weak, as it decreases the negative ET bias
against FLUXCOM by only 0.6% (1.0% for the positive Q bias against HydroPortail).

The change in the root density profile from EXP3 to EXP4 further modifies transpiration but also changes the hydraulic
conductivity of the shallow soil layers. In EXP4, we increased the root density of the crop PFTs (PFTs 12 and 13) by decreasing
$c$, and decreased the root density of the forest and boreal grass PFTs (PFTs 6, 7, 8, 9, and 15) by increasing $c$ (Table A1; Figure
F1). Given the major spatial distribution of the crop PFTs in France (Figure A1), the general effect of EXP4 compared to EXP3
increases transpiration (thus ET), which also reduces drainage at the soil bottom. In addition, the hydraulic conductivity of the
shallow soil layers in France is slightly increased but with some spatial differences, as shown in Figure D2c-d, which generally
reduces surface runoff. The negative ET bias against FLUXCOM is decreased by 1.9% (4.4% for the positive Q bias against
HydroPortail). Both EXP3 and EXP4 barely improve the correlation and KGE criteria of simulated Q against observations,
while EXP4 slightly degrades the timelag due to the change in infiltration capacity in surface soils.

In summary, by successively adjusting the surface roughness, hydraulic, and vegetation parameters, the goodness-of-fit
measures of the simulated ET and Q are gradually improved. A considerable improvement in the bias criteria for the simulated
ET and Q comes from the method of calculating ET by prescribing the surface roughness parameters (EXP1a and EXP1b). The
correlation, KGE and timelag criteria performance for the simulated Q are considerably increased by calibrating the hydraulic
parameter (EXP2) due to a better adjustment of the surface runoff and drainage ratios to total runoff. Calibrations of vegetation
parameters (EXP3 and EXP4) also improve the simulation performance but with minor sensitivity compared to the previous
calibrations. Generally, EXP4 shows the most satisfactory simulation performance among the calibration experiments.

## 3.3  Evaluation of simulated water fluxes

The simulation is evaluated using the EXP4 calibration experiment, which yields the overall best performance criterion values
in terms of the simulated ET and Q.

### 3.3.1  ET simulation performance

Both the GLEAM and FLUXCOM datasets are used to evaluate the ET simulation by ORCHIDEE in this study, and they
exhibit similar spatial patterns of ET, with more ET in the southern part (except for the high Alps) and less ET in the northern
part of the simulation domain (Figure 4a-b). However, compared with that in the GLEAM dataset, the ET of the FLUXCOM
dataset is much greater in the north-western part of the domain (i.e., the Seine and Loire River basins) and lower in the
mountainous regions (i.e., the Alps, the Massif central and the Pyrenees).



**Figure 4.** The spatial distributions of the ET datasets and the simulated ET biases against them for all the Safran grid cells over the entire simulation domain: (a) the mean ET of the GLEAM dataset and (c) the bias of the simulated ET against it from 1980 to 2020; (b) the mean ET of the FLUXCOM dataset and (d) the bias of the simulated ET against it from 2001 to 2015. The mean ET of the GLEAM dataset from 2001 to 2015 is not shown here but is very similar to that of the 1980-2020 period.





Figure 4c shows that the spatial distributions of the simulated ET biases are distinctly contrasted from those of the GLEAM dataset over the entire simulated domain: the simulated ET is generally underestimated in the mountainous regions (except for the high Alps, where a considerable overestimation occurs) and the Gascogne region (alluvial plain of the Pyrenees) but
is significantly overestimated in the north-western part (notably the Seine River basin). On the other hand, compared to the FLUXCOM dataset, Figure 4d shows that the simulated ET bias is generally underestimated but overestimated in the south-eastern part of the entire simulation domain (notably the Mediterranean mountainous regions). Although the median bias of the simulated ET against GLEAM (-0.5%) is better than that against FLUXCOM (-4.3%), as illustrated in Figure 3, the simulated ET is more spatially consistent with FLUXCOM.

**3.3.2  Q simulation performance**

Figure 5 shows the spatial distribution of the simulated Q criteria evaluated by the 1785 selected hydrometric stations in the Hy-droPortail dataset. The Q simulated by ORCHIDEE is mainly underestimated in Mediterranean river basins and overestimated elsewhere, which is consistent with the overestimation of the simulated ET in the Mediterranean region and the underestima-tion elsewhere against the FLUXCOM dataset. The biases of the simulated Q for most basins larger than $10^3$ km$^2$ are less
than 10%. In general, the river discharges in the Saône (a major tributary contributing to the Rhône River basin), Garonne, and Loire River basins along the main river networks are satisfactorily represented by ORCHIDEE, with the Pearson correlation and KGE criteria mostly larger than 0.8 and 0.75, respectively. Subbasins with areas less than $10^3$ km$^2$ in these river basins are also fairly well simulated with Pearson correlation and KGE criteria broadly larger than 0.6 and 0.5, respectively. In terms of the timelag criterion, the simulated Q of most hydrometric stations in the Seine River basin leads the observations by approxi-
mately 2 to 6 days, while the simulated Q in the Loire River basin tends to lag the observations by 2 to 4 days. The simulated Q in the Garonne and Rhône River basins generally reveal no obvious leading or lagging results.

Human impacts on water that are not explicitly taken into account in this study lead to the degradation of goodness-of-fit indicators. An example of the simulation results for the Loire River basin is illustrated in Figure 6. The variability and seasonality of the Q and ET fluxes are well represented by ORCHIDEE while the simulated Q of the Loire River during the
summer period is overestimated. This difference may be attributed to the irrigation extraction of maize crops by upstream reservoirs (Janin, 1996). The Seine River basin is influenced by upstream reservoirs that store high winter flows and release them during summer to meet environmental and navigational needs and by groundwater pumping mostly for drinking water in Paris (Flipo et al., 2020). These human interventions are not described in the ORCHIDEE LSM, which probably explains in a large part why river discharge downstream of Paris is strongly underestimated, especially in summer (Figure 7). In addition,
the mountainous basins in the Alps (e.g. the Isère and Durance River basins, contributing to the Rhône River) and the Pyrenees (e.g. the Neste River basin, contributing to the Garonne River) show unsatisfactory simulation performance (Figures G1-2) because these river basins are significantly perturbed by dams and reservoirs for winter hydropower production, spring refill and summer irrigation (e.g. the Serre-Ponçon reservoir in the Durance River basin, one of the largest dams in Europe) (François et al., 2014; Andrew and Sauquet, 2017; Huang et al., 2022; Baratgin et al., 2024). Appendix I shows that ORCHIDEE performs
better on natural or weakly influenced river basins than on influenced river basins, especially for correlation and KGE criteria.

**Figure 5.** The spatial distribution of the Q simulation performance evaluated by the (a) bias, (b) Pearson correlation coefficient, (c) KGE of the square-rooted Q, and (d) timelag for the 1785 selected French hydrometric stations in the HydroPortail dataset over the entire simulation domain.





**Figure 6.** The simulation performance of the Loire River at the hydrometric station Montjean-sur-Loire (M5300010): (a) the simulated river basin area with the legend of the proportion of Safran grids contributing to the basin area; (b) the annual regime of simulated Q compared to the observation in the HydroPortail dataset at daily time steps from 1959 to 2020; (c) the annual regime of simulated ET compared to the GLEAM dataset at monthly time steps from 1980 to 2020; (d) the annual regime of simulated ET compared to the FLUXCOM dataset at monthly time steps from 2001 to 2015; and (e) the simulated Q compared to the observation at monthly time steps from 1959 to 2020. The regime plots of ET and Q are presented with the colour bands as the range between the 25% and 75% quantiles and the solid lines as the medians.



# H8110020: Seine at Vernon



**Figure 7.** The same as in Figure 6 but for the Seine River at the hydrometric station Vernon (H8110020).





Other model imperfections degrade the quality of simulated river discharge. In the mountains, these poor results could also be related to poor snow simulation. Groundwater is also known to strongly influence streamflow, especially in river basins embedded in sedimentary basins (e.g., the Seine River basin). Groundwater is simply represented in ORCHIDEE by the slow reservoir of the routing scheme as a free aquifer. However, ORCHIDEE does not account for the difference between large

aquifers, which significantly buffer river discharge variability (Gascoin et al., 2009), and smaller aquifers with shallow and very reactive water tables. This problem could be approached by assigning larger residence times to the slow reservoirs of grid cells in sedimentary basins.

### 3.3.3 Q trend performance

Figure 8 shows that the calibrated ORCHIDEE LSM satisfactorily represents the observed Q trend of the 1785 selected French

hydrometric stations. In general, both the observed and simulated Q trends exhibit similar spatial patterns, with a decreasing trend (significant) in the south-eastern part of France and an increasing trend (not significant) in the north-western part of France, which is consistent with the findings of previous studies (e.g., Gudmundsson et al., 2017; Vicente-Serrano et al., 2019). Most basins with significant observed and simulated decreasing trends are located in the Garonne River, upstream of the Loire River, and upstream of the Rhône River. However, compared with the observed Q trends over France, the simulated Q trends

tend to alleviate the decreasing trend and to enhance the increasing trend. For example, there is a general decreasing trend, usually not significant, for the basins located on the Mediterranean coast (including Corsica) from the observed Q, while the simulated trend signals are more variant. In the north-western part of France (such as in the middle part of the Seine River and the Brittany Peninsula), the increasing trends of the simulated Q are more pronounced than in the observations. To summarize, the simulated trends tend to be more positive than the observed trends over France, which might be attributed to the fact that

intensified water withdrawals (for irrigation, industry, and drinking water) are not considered in the simulation from 1959 to 2020. This period corresponds to the expansion of agricultural and water infrastructure projects across France (Janin, 1996), and human activities are inferred to be the dominant drivers of river flow decreases and drought aggravation (e.g., García-Ruiz et al., 2011; Loon et al., 2022; Greve et al., 2023). Nonetheless, climate variability and land use and land cover changes are taken into account in the simulations of this study, which partially explain the south-east (dry)/north-west (wet) contrast across

France.

**Table 2.** The confusion matrix metrics that evaluate the performance of the ORCHIDEE LSM in representing the Q trend signal and significance over France from 1959 to 2020.

| | Trend signal | | | | |
|---|---|---|---|---|---|
| Metric | Accuracy | PPV | NPV | TPR | TNR |
| Value [-] | 76.4% | 66.8% | 81.7% | 66.7% | 81.8% |
| | Trend significance | | | | |
| Metric | Accuracy | PTSA | | NTSA | |
| Value [-] | 62.7% | 87.9% | | 79.4% | |





**Figure 8.** The spatial distribution of the observed Q trend (a), the simulated Q trend (b), and the ratio between the simulated Q trend and observed Q trend for the 1785 selected French hydrometric stations in the HydroPortail dataset from 1959 to 2020. The trends of the simulated and observed Q are calculated with the yearly mean Q time series for the common period of two time series. The confusion matrix (d) between the observed Q trend and the simulated Q trend is presented at 4 dimensions as colourful boxes (significant positive, not significant positive, not significant negative, and significant negative) and at 2 dimensions as white boxes (positive and negative).





To further advance the information obtained from the confusion matrix of Figure 8d, Table 2 summarizes the performance of the simulated Q trend signal and its significance. The classic metrics of the confusion matrix are used here including the following: the metric "Accuracy" indicates the proportions of correctly simulated trend signals and significance among the 1785 hydrometric stations; the metrics "PPV" (positive predictive value) and "NPV" (negative predictive value) indicate how many positive/negative simulations are actually correct (from Figure 8d, 423/(423+210) for PPV and 941/(941+211) for NPV); and the metrics "TPR" (true positive rate) and "TNR" (true negative rate) indicate how many positive/negative observations that are able to be correctly represented by the model (from Figure 8d, 423/(423+211) for TPR and 941/(941+210) for TNR). In addition, the metrics PTSA (positive trend significance accuracy) and NTSA (negative trend significance accuracy) indicate the proportions of accurate trend significance (either significant or not) among all the accurate positive or negative trends. Hydrometric stations with accurate trend signals but incorrect significance (in light blue in Figure 8d) are considered inaccurate in this framework.

Generally, the ORCHIDEE LSM satisfactorily reproduces the past trends of French river discharge from 1959 to 2020 with an accuracy of 76.4% for the trend signal and an accuracy of 62.7% for the trend significance. Compared to the observed Q trend signal over France, the negative Q trend is relatively better simulated than the positive Q trend despite the inadequate consideration of human perturbations. However, in terms of the simulation performance of trend significance, the positive Q trend significance is slightly better than the negative Q trend significance.

Figures 9 and 10 show the water flux trends in the Loire and Seine River basins, respectively, as two examples. The decreasing trends of the annual streamflow for both river basins are well simulated by the ORCHIDEE LSM. Nevertheless, the simulated decreasing trend of the Loire River is weakened with the underestimation of wet years, while that of the Seine River is aggravated with the overestimation of wet years and the underestimation of dry years. The simulated increasing trends of ET for both river basins are more consistent with those of the GLEAM dataset than those of the FLUXCOM dataset given its longer records. More examples of the water flux trends are provided in Appendix H.

## 4 Discussion and conclusions

This study presents the development of a high-resolution hydrological simulation over France with the ORCHIDEE LSM to quantify water resources at the nationwide scale, either retrospectively, as shown here to evaluate the set up, or prospectively, as planned for the Explore2 project to deliver climate change projections. After several calibration steps to improve the simulated water budget and hydrological performance, the simulation results over the 1959-2020 period are evaluated via comparison with the ET products (the GLEAM and FLUXCOM datasets) at monthly steps and with the French national hydrometric networks (the HydroPortail dataset) at daily time steps. Generally, the selected parametrization of the ORCHIDEE LSM provided satisfactory results in terms of the simulated basin areas and water fluxes.

This study emphasizes the ability of this version of ORCHIDEE to reproduce the temporal and spatial patterns of Q trends, with an accuracy of the trend signal of 76.4% and that of the trend signficance of 62.7% over France. The decreasing trend in south-eastern France with marked significance and the increasing trend in north-western France with minor significance are



adequately represented by the ORCHIDEE LSM. To a greater extent, this diagnosis is necessary for climate change impact
studies because adaptation strategies are grounded in the statistical analysis of the projections, potential trends of future river
discharge in particular. Therefore, hydrological models must be able to accurately reproduce trend signals under current climate
conditions. However, in most climate change impact studies, this investigation of hydrological model performance has been
rarely analysed but has been found to depend primarily on traditional goodness-of-fit indicators (e.g., KGE). A recent study
revealed that these traditional indicators do not ensure the reproduction of trend signals (Todorović et al., 2022). An inadequate
representation of vegetation dynamics can explain why some hydrological models fail to accurately reproduce river discharge
trends (e.g., the HBV conceptual hydrological model; Duethmann et al., 2020). On the other hand, the incorporation of vege-
tation dynamics in hydrological models often improves their hydrological simulation performance (e.g., Jiao et al., 2017; Bai
et al., 2018). The ORCHIDEE LSM accounts for the interactions between the biosphere (15 PFTs in this study) and hydro-
sphere (e.g., transpiration, precipitation interception, and photosynthesis) by coupling the SECHIBA and STOMATE modules
to explicitly simulate the phenomena of the terrestrial carbon and water cycles (Ducoudré et al., 1993; Krinner et al., 2005).

    Uncertainties remain in the the ORCHIDEE LSM and the selected parameterization. In this study, we applied the trial-and-
error calibration method (no parameter optimization procedure) to reduce the computational burden. The general principle of
calibration is to decrease the biases of water fluxes across France by increasing ET and decreasing Q simulations starting from
the CMIP6 configuration; the calibration experiments follow this principle by changing the parameters employed for all the
grid cells across France. This procedure indeed simplifies the calibration of such sophisticated physically based LSMs to obtain
generally accepted performance criteria over the entire simulation domain. However, improvements in simulation performance
in some areas remain limited. For example, calibrating the hydraulic conductivity influenced by both soil and vegetation (EXP2
and EXP4) to adjust infiltration and surface runoff have improved the overall performance criteria except for those of the Seine
River basin. In reality, the characteristics of the Seine River basin, such as its relief and lithology, allow more predominant
infiltration than surface runoff, especially in the upstream region of the Seine basin (e.g., Schneider et al., 2017; Mardhel
et al., 2021). The calibration of soil hydraulic conductivity at the basin scale (i.e., different parameter values for the grid cells
over the simulation domain) could improve the simulation results (e.g., Quintana Seguí et al., 2009). Another challenge is the
parameterization of the 15 PFTs in terms of their transpiration capacity when facing water stress and root profiles (EXP3 and
EXP4) due to the lack of observations, notwithstanding their importance to terrestrial carbon and water cycles.

Uncertainties concerning the input data should also be considered. For instance, heterogeneity particularly regarding the
radiation data of the Safran reanalysis (i.e., the break of homogeneity in time series with an abrupt increase in the profile of
incident solar radiation after the late 1980s) has been reported, which could be attributed to the improvement of the assimilation
system over time, the variation of the in-situ observations and the darkening-lightening effect (e.g., Le Moigne et al., 2020).
This directly impacts the ET simulation results. There are also large uncertainties in the Safran reanalysis on precipitation in
high elevation areas in France (e.g., Birman et al., 2017; Baratgin et al., 2024). The uncertainties of other input datasets, such as
the Reynolds soil texture map and the LUH2 land use and land cover maps applied to the ORCHIDEE LSM has been discussed
in other studies (e.g., Kilic et al., 2023; Lurton et al., 2020; Tafasca et al., 2020).



Nevertheless, some perspectives concerning better representations of land surface processes can be proposed to improve the simulation performance of the ORCHIDEE LSM. Given the inadequate performance of basins where drainage plays an important role in river discharge (e.g., the Seine River basins), the introduction of a groundwater module in the ORCHIDEE LSM is necessary to describe the interactions between aquifers and rivers in these basins. Moreover, mountainous basins are not adequately simulated; indeed, the Safran reanalysis is deficient in these regions, and there is still room for improvement in the 3-layer snow model, especially for the snow thermal conductivity, which is crucial for snow dynamics (Wang et al., 2013). Human impacts on terrestrial water cycles, in particular irrigation activities and dam regulations, could also be included to acquire more realistic simulations by comparison to the observations in highly anthropized basins. These functionalities are being developed by the ORCHIDEE project team at IPSL. Currently, a new irrigation module based on the flooding irrigation method (Arboleda-Obando et al., 2023) and a new demand-based hydropower module (Baratgin et al., 2024) are being developed and validated.





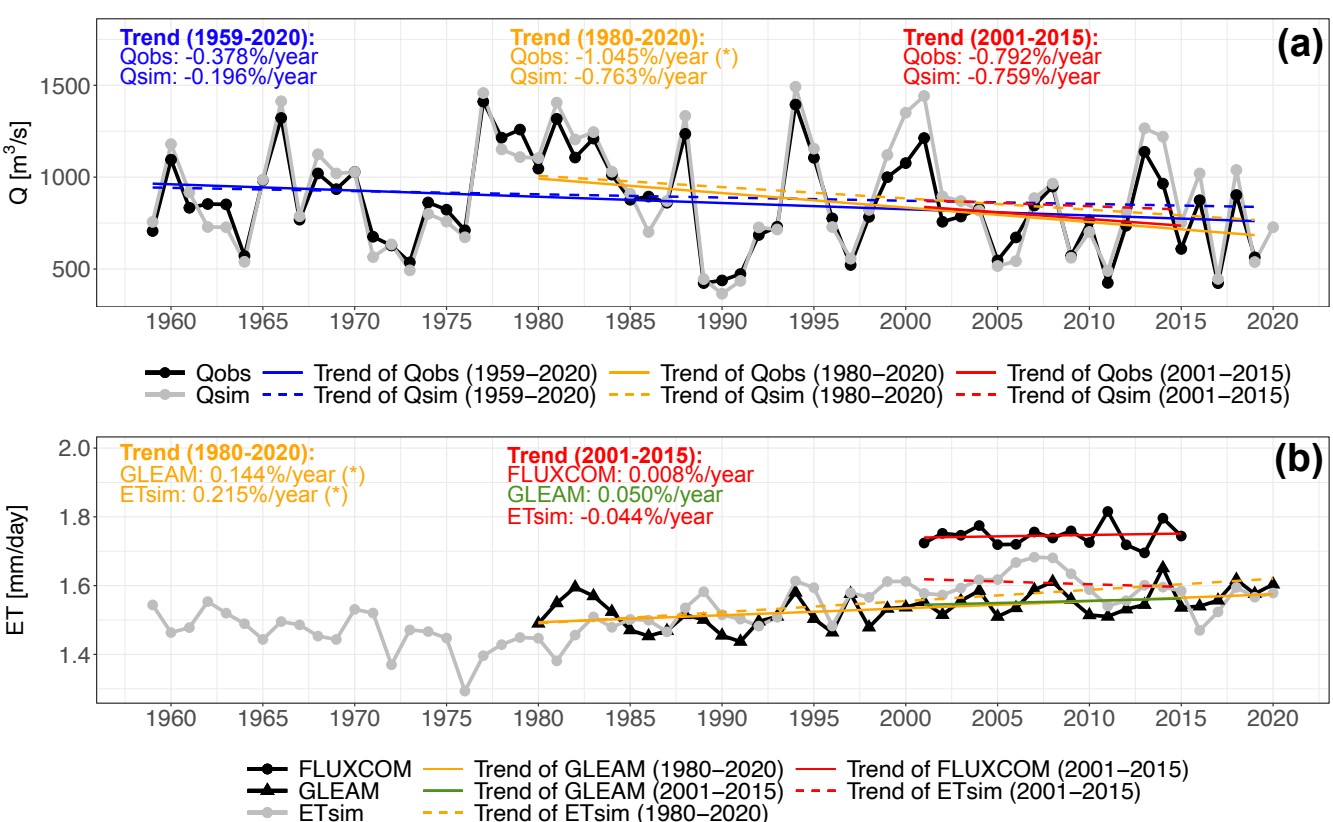

**Figure 9.** The temporal trends of the observed and simulated annual Q (a) and ET (b) of the Loire River at the hydrometric station Montjean-sur-Loire (M5300010) for the 1959-2020, 1980-2020, and 2001-2015 periods. The trend magnitudes are marked in the plots in units of %/year and the symbol (*) indicates the significance of the trend (p value <0.05).




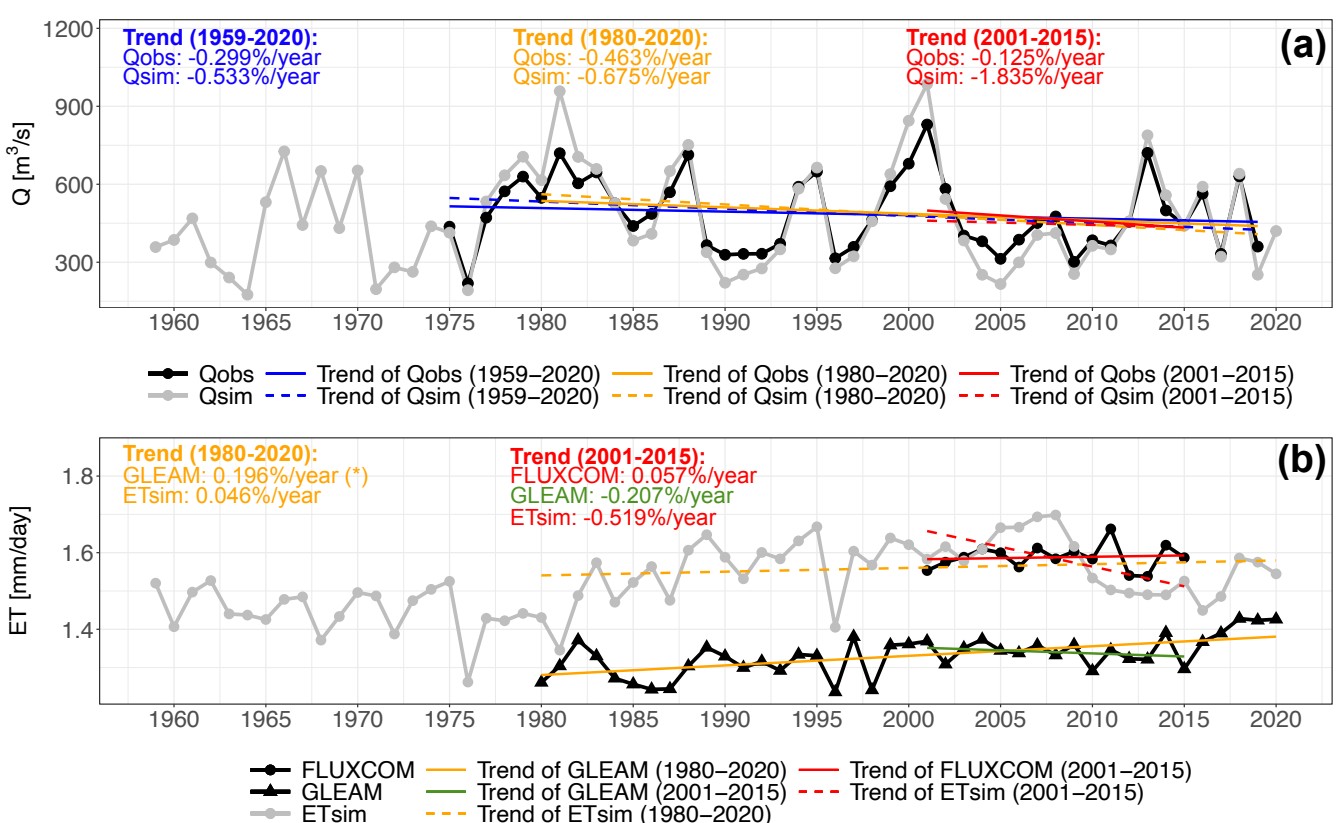

**Figure 10.** The same as in Figure 9 but for the Seine River at the hydrometric station Vernon (H8110010).





# Appendix A: PFTs in the ORCHIDEE LSM

**Table A1.** Overview of the 15 PFTs used in ORCHIDEE with the root density value $c$ of each PFT used for CMIP6 configuration (ref) and for this study (new). PFTs 2, 3, and 14 do not appear in France.

| PFT | Description | $c$ of ref [-] | $c$ of new [-] |
|-----|-------------|---------------|---------------|
| 1 | bare soil | 5.0 | 5.0 |
| 2 | tropical broadleaf evergreen | 0.8 | 0.8 |
| 3 | tropical broadleaf raingreen | 0.8 | 0.8 |
| 4 | temperate needleleaf evergreen | 1.0 | 1.0 |
| 5 | temperate broadleaf evergreen | 0.8 | 0.8 |
| 6 | temperate broadleaf summergreen | 0.8 | 1.5 |
| 7 | boreal needleleaf evergreen | 1.0 | 2.0 |
| 8 | boreal broadleaf summergreen | 1.0 | 2.0 |
| 9 | boreal needleleaf deciduous | 0.8 | 1.5 |
| 10 | temperate natural grassland (C3) | 4.0 | 4.0 |
| 11 | natural grassland (C4) | 4.0 | 4.0 |
| 12 | crops (C3) | 4.0 | 2.0 |
| 13 | crops (C4) | 4.0 | 2.0 |
| 14 | tropical natural grassland (C3) | 4.0 | 4.0 |
| 15 | boreal natural grassland (C3) | 4.0 | 6.0 |



**Figure A1.** The spatial distribution of the PFTs that appear in France averaged over the 1959-2020 period based on the LUH2 dataset (Hurtt et al., 2020) reaggregated into Safran grid cells.





**Appendix B: The Reynolds soil texture map**

**Table B1.** Overview of the 12 soil texture classes in the Reynolds soil texture map (Sa = sand, Si = silt, L = loam, and C = clay) (Reynolds et al., 2000), their reference hydraulic conductivity ($K_s^{\text{ref}}$) and their water retention properties (volumetric water content at saturation $\theta_s$, field capacity $\theta_c$ and wilting point $\theta_w$ ).

| Index | 1 | 2 | 3 | 4 | 5 | 6 | 7 | 8 | 9 | 10 | 11 | 12 |
|---|---|---|---|---|---|---|---|---|---|---|---|---|
| Soil texture | Sa | LSa | SaL | SiL | Si | L | SaCL | SiCL | CL | SaC | SiC | C |
| $K_s^{\text{ref}}$ [mm/d] | 7128.0 | 3501.6 | 1060.8 | 108.0 | 60.0 | 249.6 | 314.4 | 16.8 | 62.4 | 28.8 | 4.8 | 48.0 |
| $\theta_s$ [m$^3$/m$^3$] | 0.43 | 0.41 | 0.41 | 0.45 | 0.46 | 0.43 | 0.39 | 0.43 | 0.41 | 0.38 | 0.36 | 0.38 |
| $\theta_c$ [m$^3$/m$^3$] | 0.0493 | 0.0710 | 0.1218 | 0.2402 | 0.2582 | 0.1654 | 0.1695 | 0.3383 | 0.2697 | 0.2672 | 0.3370 | 0.3469 |
| $\theta_w$ [m$^3$/m$^3$] | 0.0450 | 0.0570 | 0.0657 | 0.1039 | 0.0901 | 0.0884 | 0.1112 | 0.1967 | 0.1496 | 0.1704 | 0.2665 | 0.2707 |





**Figure B1.** The dominant soil texture classes in France based on the Reynolds soil texture map (Reynolds et al., 2000) reaggregated into Safran grid cells.



## Appendix C:  Calibration of roughness length parameters

**Figure C1.** The spatial distributions of the simulated $z_{0h}$, $z_{0m}$, and ET values averaged over the 1959-2020 period in France for the calibration experiments STD, EXP1a, and EXP1b.



**Figure C2.** The time series of the simulated $z_{0h}$, $z_{0m}$, and ET values averaged over France from 1959 to 2020 for the calibration experiments STD, EXP1a, and EXP1b.





## Appendix D: Calibration of soil hydraulic conductivity



**Figure D1.** Two examples showing how soil hydraulic conductivity is impacted by soil compaction (an exponential decrease with depth) and vegetal roots (an increase towards the soil surface): the loamy soil class (L) covered by trees (PFT6) in the left and by crops (PFT12) on the right for the calibration experiments EXP1b, EXP2 (or EXP3), and EXP4.



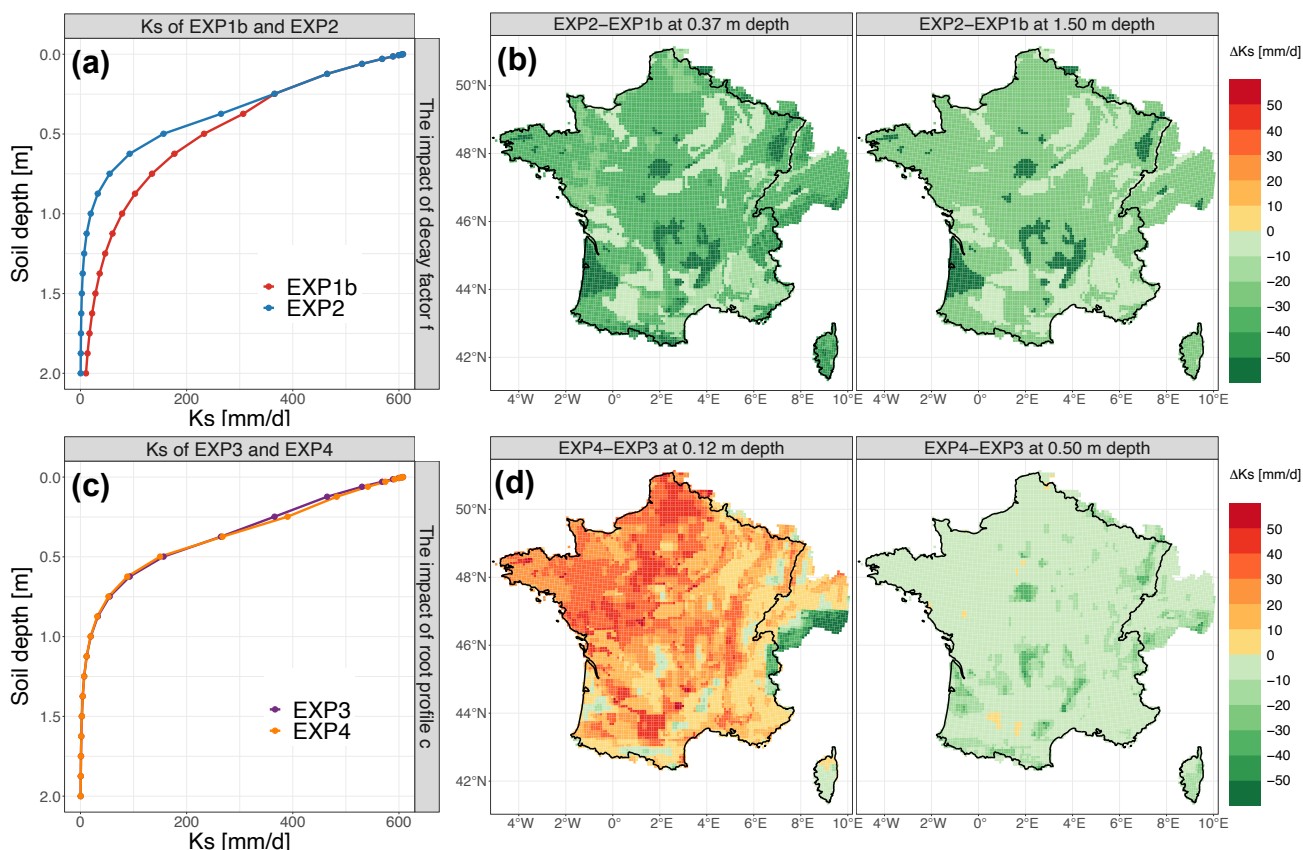

**Figure D2.** The simulated hydraulic conductivity in all soil layers averaged over France for the comparison between EXP1b and EXP2 (a) and the comparison between EXP3 and EXP4 (c). The difference in the simulated hydraulic conductivity between EXP1b and EXP2 (b) and between EXP3 and EXP4 (d) in some soil layers in France.





**Appendix E:  Calibration of the soil water stress for transpiration**

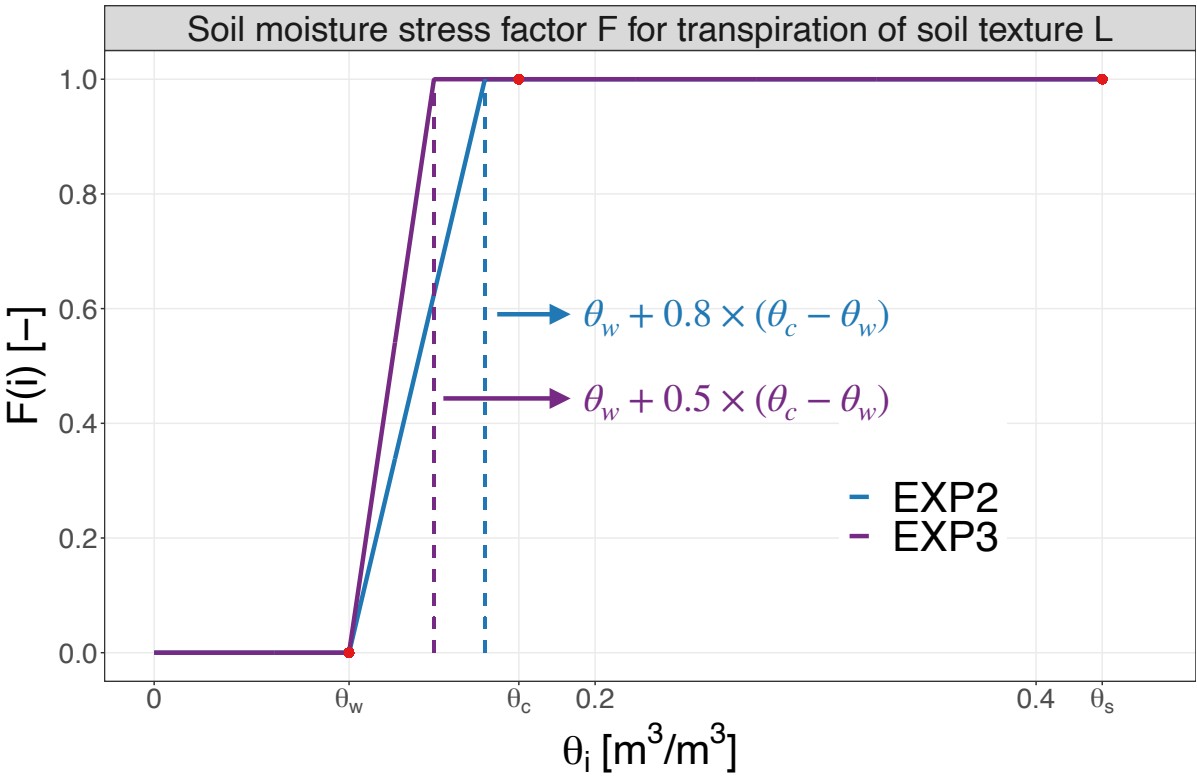

**Figure E1.** An example of the loamy soil texture (L) at soil depth layer i to show how soil moisture linearly constrains transpiration for calibration experiments EXP2 and EXP3. The water retention properties of the loamy soil texture are sourced from Table B1. When the soil moisture stress factor F is 0, transpiration is 0, and when F is 1, it reaches a maximum. F is 1 when the soil moisture is above $\theta_w + p \times (\theta_c - \theta_w)$ with $p$ equal to 0.8 for EXP2 and 0.5 for EXP3.



**Figure E2.** The transpiration factor (Us) at the PFT level averaged over France and the 1959-2020 period for the calibration experiments EXP2 and EXP3 in the top, and the difference in the bottom.



## Appendix F: Calibration of root density

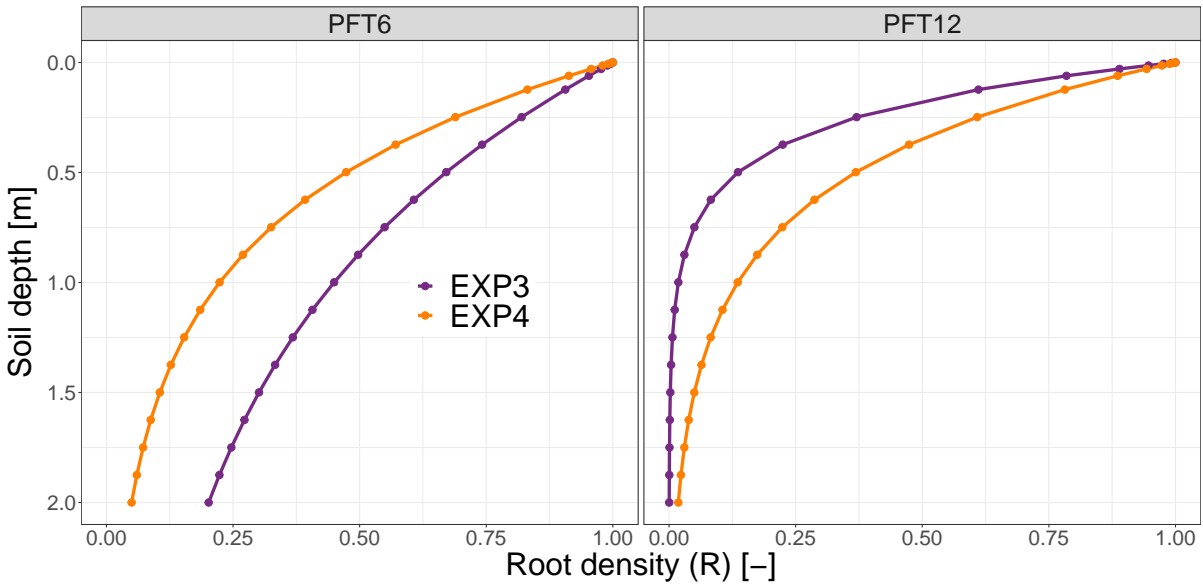

**Figure F1.** Two examples, PFTs 6 and 12, are used to show how root density changes with soil depth for calibration experiments EXP3 and EXP4.



**Appendix G:  The simulation performance of the ORCHIDEE LSM for several large river basins**

**Figure G1.** The same as in Figure 6 but for the Garonne River at the hydrometric station Lamagistère (O6140010).



# V7200010: Rhône at Beaucaire

Figure G2. The same as in Figure 6 but for the Rhône River at the hydrometric station Beaucaire (V7200010).



**Appendix H: The simulated trends of water fluxes for several large river basins**

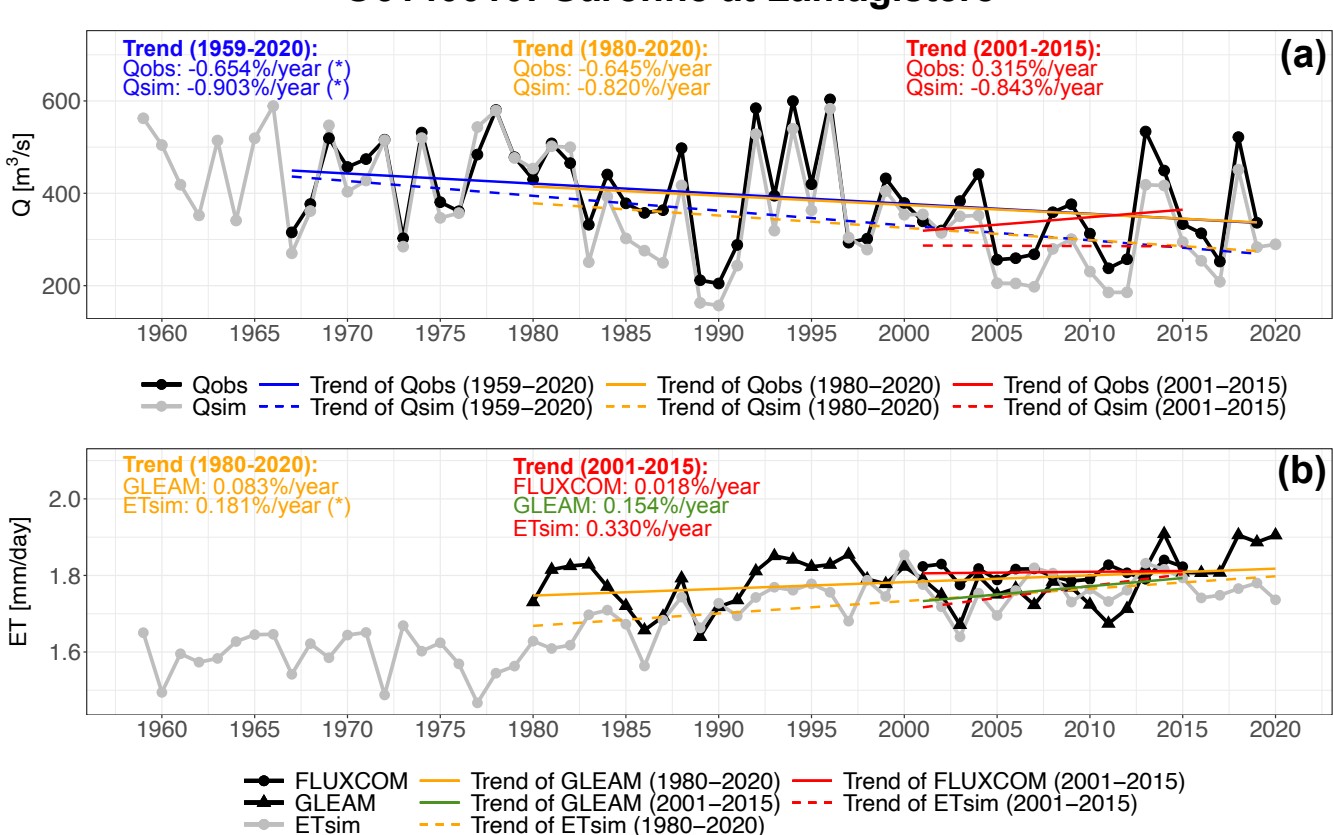

**Figure H1.** The same as in Figure 9 but for the Garonne River at the hydrometric station Lamagistère (O6140010).



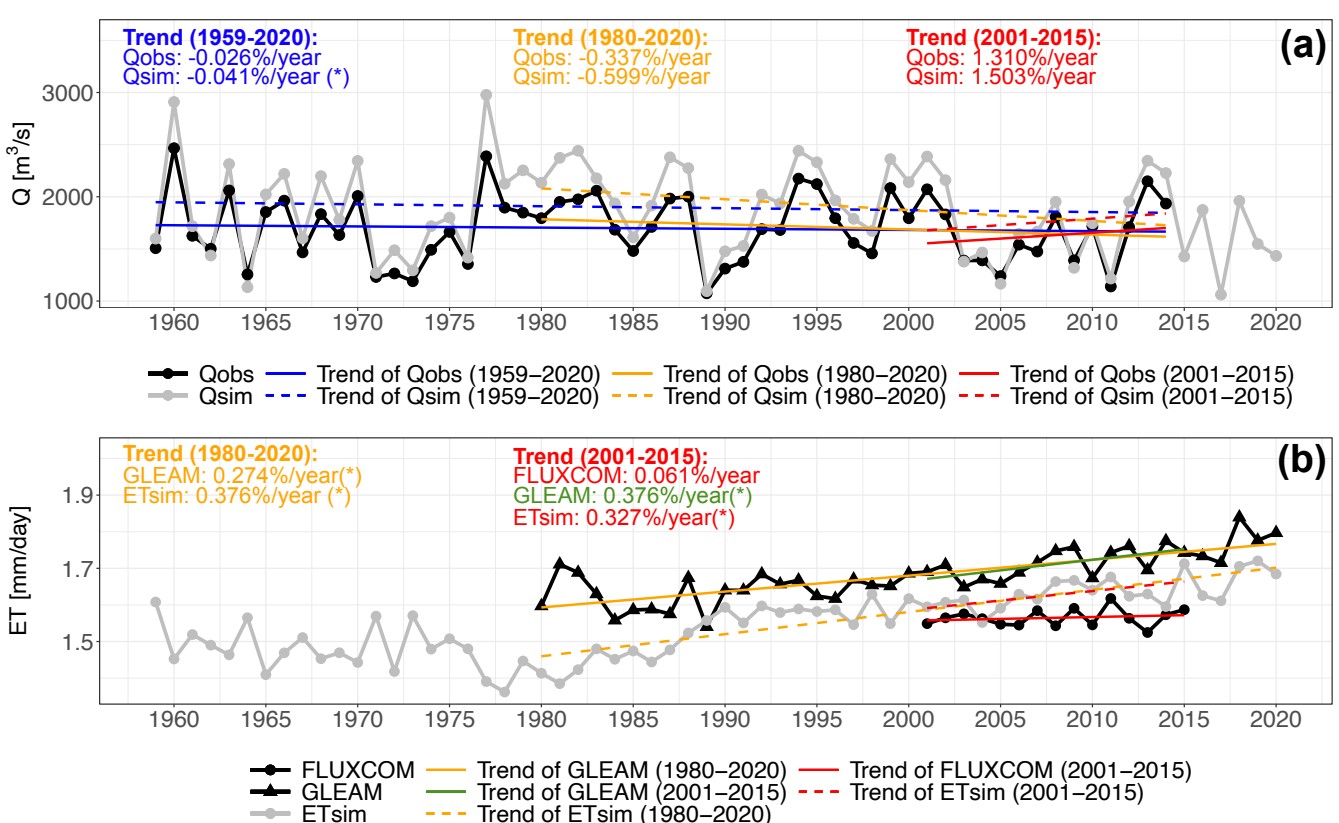

**Figure H2.** The same as in Figure 9 but for the Rhône River at the hydrometric station Beaucaire (V7200010).



## Appendix I: Human impacts on the simulated Q performance

Within the framework of the Explore2 project, the identification of human impacts on French river basins was based on the procedure of cross-referencing the Q observations from HydroPortail and possible sources of human activities on natural water

resources. These sources are the French national BNPE dataset (https://bnpe.eaufrance.fr) that records water withdrawals as well as their uses for all the communes in France, and the AQUASTAT dataset (www.fao.org/aquastat/en/databases/dams that contains descriptive data (location, volume, and function) for 113 large dams over France. In addition, several indicators are calculated to quantify the human impacts on water resources: the ratio of upstream reservoir volume to naturalized annual Q, the ratio of summer withdrawals to naturalized summer Q, and the ratio of annual withdrawals to naturalized annual Q. The

values of these indicators for natural or weakly influenced basins should be less than 5%, 10%, and 5%, respectively. The basins that do not meet the criteria are identified as influenced. Figure I1 shows that ORCHIDEE simulates Q better for the natural or weakly influenced basins than the influenced basins.

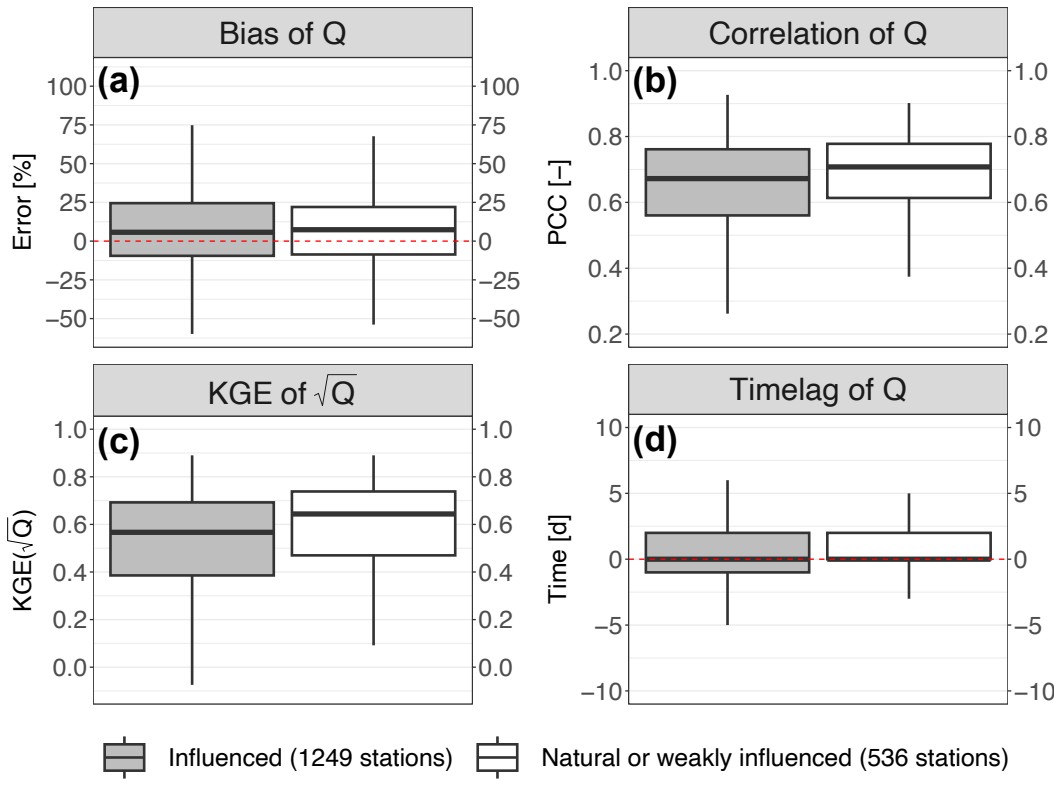

**Figure I1.** Performance criteria of the simulated Q in EXP4 for the influenced basins compared to the natural or weakly influenced basins: (a) bias; (b) Pearson correlation coefficient; (c) KGE of square-rooted simulated Q; (d) timelag.



*Code and data availability.* The ORCHIDEE codes and data used in this study can be obtained by contacting the corresponding author.

*Author contributions.* PH, AD, and LR designed the research. PH and LR performed the simulations. PH analysed the data and prepared a
draft of the manuscript. JP and LB contributed to building the high-resolution river network. VB prepared the input data for the simulation.
ES prepared the river discharge observations. All the authors contributed to interpreting results, discussing findings and improving the
manuscript.

*Competing interests.* The authors declare that they have no conflict of interest.

*Acknowledgements.* This research was financed by the Explore2 project with support from the French Biodiversity Agency (OFB) and the
French Ministry of Ecological Transition (MTECT). This research was also financed by the BLUEGEM project (ANR-21-SOIL-0001). The
simulations analyzed in the paper were produced and stored at THE IDRIS on the supercompute Jean Zay CSL, owing to THE resources
provided by GENCI under grant 2022-AD010113599.



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
