# Peer review of "Multi-objective calibration and evaluation of the ORCHIDEE land surface model over France at high resolution"

_EGUsphere, 2024_

## Author Comment (AC1)

**Response to the comments of anonymous referee #1**

We would like to thank anonymous referee #1 for the comments that help to improve the manuscript. Below are our responses to the concerns raised. The comments of anonymous referee #1 are shown in black. Authors' responses are shown in blue.

Summary
The authors present multi-objective calibration and evaluation for high-resolution hydrologic simulations over France produced by the ORCHIDEE land surface model (LSM). They conduct a comprehensive evaluation for the model performance, considering both classic goodness-of-fit indicators including KGE and bias and trends in streamflow and ET. The comparison is promising. Overall, the manuscript is well-written with high-quality figures.
However, I still have the following concerns.

Response: We thank this positive evaluation of our work. We are also thankful for the constructive comments and suggestions provided which have certainly helped improve the paper.

1. In the abstract, the authors claim that they present a strategy to obtain a realistic hydrological simulation over France, but in fact, the ORCHIDEE land surface model does not consider human impacts, leading to poor performance in some regions in France. Therefore, it would be more precise to state "a reliable hydrological simulation".

Response: We thank anonymous referee #1 for this suggestion. Following this suggestion, we will change "realistic" to "reliable".

2. Through the comparison of streamflow and ET, the model always performs better against GLEAM than FLUXCOM. Is that because GLEAM reanalysis data is still modeling data and FLUXCOM is generated based on observational datasets? If the ORCHIDEE land surface model uses similar physics equations to those of GLEAM, we may expect the results from ORCHIDEE and GLEAM are in good agreement.

Response: There are no perfect evapotranspiration products given the inherent uncertainties (e.g., Liu et al., 2023; Xie et al., 2024). Both GLEAM and FLUXCOM products are based on sound methodologies: GLEAM generates ET at 0.25° resolution from 1980 to 2020 based on the Priestley-Taylor potential evapotranspiration formula with satellite-based products (net radiation, precipitation, surface soil moisture, skin and air temperatures, vegetation optical depth, and snow water equivalent) as inputs; FLUXCOM relies on machine learning algorithms to spatially interpolate in situ FLUXNET measurements at 0.5° resolution from 2001 to 2015, using constraints from remote sensing and meteorological observations.

ORCHIDEE calculates ET as the sum of plant transpiration, evaporation of intercepted water, soil evaporation and snow sublimation based on water and energy budgets.

We can consider, however, that ORCHIDEE, GLEAM and FLUXCOM provide independent estimates of ET, all with their own uncertainties (Liu et al., 2023). In this framework, our guideline was to compare the ORCHIDEE simulation with several products, which offers plausible range. Eventually, in our study, we do not conclude that "the model always performs better against GLEAM than FLUXCOM": the bias of the simulated ET to GLEAM over the entire study domain is better than that to FLUXCOM, but the simulated ET is more spatially consistent with FLUXCOM than GLEAM (L302-309 and Figure 4).

To be clear, we did not stop at EXP4 because it has a good bias value compared with GLEAM (-0.5%). Instead, ET is underestimated compared with FLUXCOM (-4.3%) and Q is overestimated compared with observations (6.3%), which means that there is probably a physical consistency between these two datasets. We simulated the natural behavior of the French water system without considering human perturbations, such as pumping and irrigation, which could result in an underestimation of ET and an overestimation of Q if we consider FLUXCOM and Q observations at the same time. Anyway, we did not seek to obtain perfect bais values against these datasets but we try to make some compromises to make our manual calibration more reasonable and reliable.

Following this comment, we will add a sentence in L203: "GLEAM and FLUXCOM provide independent ET estimates, both of them with large uncertainties (Liu et al., 2023). They are used in combination to approach the plausible range of observed ET."

3. In the Introduction section (Line 77-87, Page 3), the authors introduce the first distributed LSM at the nationwide scale of France, SIM. SIM has shown very good performance in generating hydrologic simulations. Why do the authors decide to use another LSM, ORCHIDEE for France? What are the limitations of SIM?

Response: There are no perfect models for hydrological simulations and each model has its own strengths and limitations. It is always encouraging to have several models that are capable of obtaining reliable hydrological simulations at the nationwide scale of France, especially when the goal is to test the response to changing conditions (e.g. climate change, land use change, etc.) . Multi-model assessment is more robust than single one model for hydrological simulation and projection. Both SIM and ORCHIDEE LSMs have contributed to the national EXPLORE2 project (https://professionnels.ofb.fr/fr/node/1244) for climate change impact analysis.

4. In Line 264-265, Page 11, the authors state that "The timelag criterion of the simulated Q is also greatly improved from a range of -11 to 27 days to a range of -3 to 5 days.". In my opinion, the zero timelag is the best, right?

Response: Yes, and we will underline this in L211 of the paper for the sake of clarity.

5. Specific comments: Line 102, Page 4: "(revision 7738)" should be deleted.

Response: Thanks for the suggestion. This will be deleted from the title, but for clear documentation of the code (refer to EGU guidelines), we will change L103-105 to "The ORCHIDEE model is a physically-based LSM developed at the Institut Pierre Simon Laplace (IPSL) as the land component of the IPSL climate model, which is used for all the past and future climate simulation exercises carried out for the IPCC reports as part of the Coupled Model Intercomparison Project (CMIP) (IPCC,2023). Here, we use ORCHIDEE version 2.2 (with revision 7738), which is very close to the version used as the land component of the IPSL-CM6 climate model (Boucher et al., 2020; Cheruy et al., 2020)."

6. Line 219, Page 8: "STD" first appears. What is "STD"?

Response: We will reconstruct the sentence in L219 to "The starting point experiment of calibration design is called STD and uses the "standard" parameter set sourced from CMIP6…".

7. Table 2: Please explain the meanings of the labels in the caption, such as "PPV".

Response: Thank you for this suggestion and for the sake of clarity, we will explain the meanings of the labels in section 2.3.

**Final note: We will also correct the style of some sentences so that they are more readable, and orthographic and grammar errors.**

**References**
Boucher et al. (2020), Presentation and Evaluation of the IPSL-CM6A-LR Climate Model, J. Adv. Model. Earth Syst., 12, e2019MS002010, doi: 10.1029/2019MS002010.

Cheruy et al. (2020), Improved Near-Surface Continental Climate in IPSL-CM6A-LR by Combined Evolutions of Atmospheric and Land Surface Physics, J. Adv. Model. Earth Syst., 12, e2019MS002005, doi: 10.1029/2019MS002005.

IPCC (2023), Climate Change 2021 - The Physical Science Basis: Working Group I Contribution to the Sixth Assessment Report of the Intergovernmental Panel on Climate Change, Cambridge University Press, doi: 10.1017/9781009157896.

Liu et al. (2023), Intercomparison and evaluation of ten global ET products at site and basin scale, J. Hydrol., 617, 128887, doi:10.1016/j.jhydrol.2022.128887.

Xie et al. (2024), Evaluation of seven satellite-based and two reanalysis global terrestrial evapotranspiration products, J. Hydrol., 630, 130649, doi:10.1016/j.jhydrol.2024.130649.

---

## Author Comment (AC2)

**Response to the comments of anonymous referee #2**

We would like to thank anonymous referee#2 for the comments that help to improve the manuscript. Below are our responses to the concerns raised. The comments of anonymous referee #2 are shown in black. Authors' responses are shown in blue.

Summary
The article presents the calibration and the evaluation of the ORCHIEE land-surface model over France. The article has several major limitations. Several important choices of the methodology applied by the authors are not well explained or justified. The model version with modified parameters sets provides less biased results than the standard version, but it is difficult to evaluate whether the results should be considered as satisfactory over the test domain due to the lack of external benchmark. Besides some explanations on model failure remain unverified hypotheses.

Response: We are thankful for the thorough comments and constructive suggestions provided in this review. We kindly appreciate the efforts made by referee#2.

Major comments
1. Title: As detailed in comments below, I think the article does not explain how the "multi-objective calibration" of the model was done (or at least I did not understand that). This is a strong limitation of the article.

Response: This study aims to use the ORCHIDEE land surface model to obtain a reliable hydrological simulation over France at high resolution. To achieve this, we applied a multi-objective calibration, which means that we reduced both simulated ET and Q biases as different objectives with the help of two ET products and Q observations to constrain parameters. Then, we need to find a compromise between these different objectives to define an acceptable parameter set.

The term "multi-objective" is mostly found in automatic calibration studies of hydrological modeling that apply optimization techniques to minimize/maximize a multi-objective function. And we think referee#2 might confuse our study with these studies because there are many comments below focusing on how we optimize in the calibration procedure.

In this work, we did not use optimization techniques to calibrate ORCHIDEE. Instead, we progressively changed our parameter set to improve the overall performance of our models, as assessed via performance indicators, estimated in many spatial elements (N grid cells and Y discharge stations). In doing so, we rely on our own expert judgment to make the tradeoffs between sometimes contrasting criteria (e.g. opposite biases of ET with GLEAM and FLUXCOM). This may seem

subjective, but multi-objective optimization methods are often ineffective in such situations anyway (Vrugt et al., 2003; Chiandussi et al., 2012).

To avoid the confusion of future readers, we will explain this in Section 2.4 "Calibration design" (see the response of the comment N°11).

2. L10. Getting an almost perfect match on actual evapotranspiration given the uncertainty on the observational product used may be considered as overcalibration.

Response: To be clear, we did not stop at EXP4 because it has a good bias value compared with GLEAM (-0.5%). Instead, ET is underestimated compared with FLUXCOM (-4.3%) and Q is overestimated compared with observations (6.3%), which means that there is probably a physical consistency between these two datasets. We simulated the natural behavior of the French water system without considering human perturbations, such as pumping and irrigation, which could result in an underestimation of ET and an overestimation of Q if we consider FLUXCOM and Q observations at the same time. Anyway, we did not seek to obtain perfect bais values against these datasets but we try to make some compromises to make our manual calibration more reasonable and reliable.

This almost perfect bias value actually hides a large spatial variability over the study domain.

Response: Following this comment, the sentences in L10-13 will be changed to "For example, the median bias of evapotranspiration is -0.5% against GLEAM (-4.3% against FLUXCOM), the median bias of river discharge is 6.3%, and the median KGE of square-rooted river discharge is 0.59. These indicators, however, exhibit a large spatial variability, with poor performance in the Alps and the Seine sedimentary basin. The spatial contrasts and temporal trends of river discharge across France are well represented with an accuracy of 76.4% for the trend sign and an accuracy of 62.7% for the trend significance."

3. L10-15. I found these sentences too optimistic on model results. It seems that there are many modelling problems remaining to well capture the actual hydrological dynamics. I did not see convincing demonstration that these results would provide a "thorough historical overview of water resources" nor a "robust configuration for climate change impact analysis". The study does not analyse how model performance evolves over the study period and the model robustness is not evaluated by dedicated tests (model robustness to extrapolate in space or time).

Response: We didn't conduct split sample tests in our study because we think split sample tests are used to verify the robustness of parameters in the optimization procedure, which is not suitable in the context of manual calibration (our study).

However, we have conducted trend analyses over France, which we consider more suitable in our case to verify the robustness of the model.

How model robustness evolves over the study period will be discussed in comment N°12. We still provided the results of split sample tests in comment N°12 for your information.

In terms of "thorough historical overview of water resources", we agree with referee #2 that the description might be too optimistic and the sentence of L13-15 will be changed to "Although it does not yet integrate human impacts on river basins, the selected parameterization of ORCHIDEE offers a reliable historical overview of water resources and a robust configuration for climate change impact analysis at the nationwide scale of France."

4. L65-67: Though erroneous data may prevent obtaining good calibration results, the model itself is generally the main problem in getting good results.

Response: Yes. Based on this suggestion, we will change the sentence to "Either way, a perfect calibration is always impossible to achieve due to inherent uncertainties in forcing data (e.g., Gelati et al., 2018; Kabir et al., 2022), benchmark observations (e.g., Zeng and Cai, 2018) and model structure (e.g., van Kempen et al., 2021)."

5. L120: Why soil is 2-m deep everywhere? Is not that a strong approximation given that it is not the case?

Response: If we compare it to the reality, it is a strong approximation and simplification because soil depth is different everywhere. In land surface modeling, we use this approximation due to the lack of soil depth maps with good quality.

6. L126: Why 22 layers? Are they all of the same depth? Is this level of complexity justified by model performance (and consistent with the approximation mentioned in the previous comment)?

Response: The 2-m soil in ORCHIDEE is discretized into 22 layers (increasing soil thickness above 0.2 m and constant soil thickness below 0.2 m) to describe the vertical soil water fluxes calculated with Richards equations. The depth at which constant layer thickness starts (0.2 m here) is a parameter for soil discretization in ORCHIDEE.

The thickness of the 22 layers used in this study are shown in the table below.

| Layer | 1 | 2 | 3 | 4 | 5 | 6 | 7 | 8 |
|---|---|---|---|---|---|---|---|---|
| Thickness (m) | 0.0010 | 0.0029 | 0.0059 | 0.0117 | 0.0235 | 0.0469 | 0.0938 | 0.1251 |

| Layer | 9 | 10 | 11 | 12 | 13 | 14 | 15 | 16 |
|---|---|---|---|---|---|---|---|---|
| Thickness (m) | 0.1251 | 0.1251 | 0.1251 | 0.1251 | 0.1251 | 0.1251 | 0.1251 | 0.1251 |

| Layer | 17 | 18 | 19 | 20 | 21 | 22 | | |
|---|---|---|---|---|---|---|---|---|
| Thickness (m) | 0.1251 | 0.1251 | 0.1251 | 0.1251 | 0.1251 | 0.0626 | | |

Campoy et al. (2013) showed that a refined soil layers increased the performance of ORCHIDEE in simulating vertical water fluxes.

Following this comment, we will change the sentence in L125-128 to "At each time step, soil moisture is redistributed vertically according to the Richards equation (flow in an unsaturated medium) taking into account surface boundary conditions by infiltration and soil evaporation, withdrawals by roots through the entire soil depth to supply transpiration, and gravitational drainage at the bottom of the soil (Campoy et al., 2013; Tafasca et al., 2020). For accurate computation, soil moisture and vertical water fluxes are discretized across 22 layers over 2 m, with 7 soil layers of increasing depth in the top 20 cm to capture the strong soil moisture gradients, then soil layers of constant thickness (12.5 cm) down to the soil bottom (Campoy et al., 2013)."

7. L147-156: Though linear stores are very commonly used in hydrological modelling, they have limited efficiency in simulating some flow ranges, typically low flows. Why only linear stores are used in the model?

Response: It is true that the linear reservoir method is a strong simplification of river flow physics, but it is still widely used for routing lateral water flows in land surface models, such as ORCHIDEE (used in this study) and CLM (Lawrence et al., 2011).

The linear reservoir method reduces computational cost, especially for high-resolution application (Sheng et al., 2017; Nguyen-Quang et al., 2018; Polcher et al., 2023).

8. Section 2.3: It was not fully clear for me at which time steps the evaluation criteria were calculated. At the monthly time step for ET and daily time step for Q? The authors could also explain which expected model qualities are assessed by the criteria selected. Especially, one could expect that there could be some criteria focusing on high and low flows. Bias and correlation coefficient are two of the three components of the KGE criterion. Would the third component (ratio of variances or ratio of variation coefficient) be useful to consider also?

Response: In this study, ET was evaluated at monthly time step and Q was evaluated at daily time step (as stated in L202-204 and L207-208).

We will add a sentence in L208 "The square-root transformation on Q is applied to calculating KGE criterion to capture both high and low flows (Song et al., 2019)."

To keep the conciseness of the article, the third component of KGE criterion (variability ratio) is not explicitly demonstrated but included in the evaluation of the KGE criterion on square-rooted Q.

9. Section 2.3: It is unclear whether the observed time series were visually checked before use.

Response: The observed Q time series were not fully visually checked before use given the thousands of stations collected in the project.

In large datasets, there are often many remaining observational errors, which may strongly influence model evaluation.

Response:  Thank you for point out this and we will add this aspect in L427 of Section 4: "Besides, Q records of the selected 1785 stations are not fully checked because identifying Q anomalies is extremely time-consuming and subjective. The remaining observational Q errors might influence model evaluation."

10. L205: One part of the evaluation is on trends. However 8 years (for the shortest series) are too short to evaluate trends. This evaluation should be restricted to stations where there are long time series (at least 30 years).

Response: The table below shows the trend accuracy (EXP4) is not sensitive to the length of evaluation period.

| Period length | 8 | 15 | 20 | 25 | 30 |
|---|---|---|---|---|---|
| Number of stations | 1785 | 1477 | 1306 | 1106 | 969 |
| Proportion to 1785 stations | 100% | 83% | 73% | 62% | 54% |
| Trend signal accuracy | 76.4% | 76.0% | 76.0% | 76.7% | 78.3% |
| Trend significance accuracy | 62.7% | 61.9% | 61.0% | 60.6% | 60.9% |

Besides, the figure below shows that the performance criteria on Q are not sensitive to the length of evaluation period either.

[Figure]

Therefore, since the trend results and the performance criteria are not sensitive to the Q data length, we chose the Q records with at least 8 entire years to maximize the coverage of French rivers.

We will add a sentence in L206: "Q records with at least 8 entire years were chosen because (1) the simulation performance was shown to be rather insensitive to the length of evaluation period above 8 years (see Section S10 in supplementary material), and (2) these stations offer a large coverage of French rivers."

We will also add this information in supplementary material.

11. Section 2.4: This section is not detailed enough to fully understand what was done by the authors.

Response: Based on the general comment here and specific comments below, we will enrich Section 2.4 by changing L216-222 to:

"LSMs are complex models, integrating many coupled processes related to hydrology, soils, vegetation, but also to the radiative transfer or the boundary layer. They are also distributed models designed to be applied over wide and contrasted domains (Clark et al., 2015), in which every grid-point could be regarded as one 1D model. Therefore, their calibration is challenging and a full optimization of LSM's parameters is practically intractable, due to the computational burden (Bierkens et al., 2015), the equifinality (Fisher and Koven, 2020), and the uncertainty of input and benchmark datasets (Best et al., 2015). The most common solutions are to rely on transfer functions to derive the spatial variations of model parameters from maps of physical parameters (e.g., Samaniego et al., 2010), and to accept sub-optimal but satisfactory performance (Best et al., 2015).

In this line, we chose here to calibrate selected parameters of the ORCHIDEE LSM based on an iterative trial-and-error procedure to gradually improve simulations by manually adjusting some parameters. The simulation period is from 1959 to 2020, with a warm-up from 1959 to 1968 to provide reasonable initial conditions, and the output variables (e.g., ET and Q) are aggregated to daily time steps. The starting experiment of this calibration procedure is called STD and uses the "standard" parameter set sourced from CMIP6 (Boucher et al., 2020), according to which the roughness heights z0m and z0h are calculated by the dynamic method of Su et al. (2001). STD forced with Safran reanalysis significantly underestimates ET and overestimates Q compared with the evaluation datasets detailed in Section 2.3. The calibration is therefore aimed at increasing ET and decreasing Q. According to expert knowledge on the parameter sensitivity of ORCHIDEE and previous calibration exercises (Kiliç et al., 2023; Raoult et al., 2021; MacBean et al., 2020; Dantec-Nedelec et al., 2017; Campoy et al., 2013), we focused on parameters that control surface roughness, soil hydraulics and vegetation morphology (detailed in Section 2.1) to improve the simulations of ET and Q.

A hundred parameter sets were tested in this iterative evaluation process, summarized in Table 1 by a selection of 6 calibration experiments that show a gradual decrease of ET and Q biases on average over France. Each parameter set in Table 1 is applied uniformly over the entire simulation area."

The point-to-point response to all the specific comments concerning the calibration in our study are provided as follows.

Table 1 seems to suggest that 5 parameters were selected and that only six combinations of these parameters were tested. Is that what was done? If yes, I do not think this can be considered an actual calibration.

Response: We tested around 100 different parameter sets, some of them not selected here. We present only the most sensitive parameter changes, according to

their own physical basis, previous calibration work of ORCHIDEE (e.g., Kiliç et al., 2023; Raoult et al., 2021; MacBean et al., 2020; Dantec-Nédélec et al., 2017), and our own subjective analysis.

The calibration in our study is not based on optimization, but in the sense that we change our parameters (parameter tuning) to improve the simulation compared to the selected evaluation datasets.

Calibration is generally understood as a search of an optimum in the multi-dimensional parameter space. Here one cannot say that testing six parameter sets is an actual search. If the authors came to these values after a search in the parameter space, the way this search was done should be explained.

Response: As we explained in the introduction (L56-67), the calibration of land surface models is challenging given their hundreds of parameters (more than 500 parameters in ORCHIDEE). The major limitation of optimization-based calibration is computational burden, especially for high-resolution applications (662 hours required to run one parameter set over the 1959-2020 period in our study and more than 10 hours with a high performance computer of 64 cores), and equifinality issue. Even calibrating the selected parameters in Table 1 (local optimum search) faces the same problems.

The most common solutions are to rely on transfer functions to derive the spatial variations of model parameters from maps of physical parameters (e.g., Samaniego et al., 2010), and to accept sub-optimal but satisfactory performance (Best et al., 2015).

In this study, we manually calibrated ORCHIDEE with trial-and-error procedures (no optimization) based on expertise (L217-218). Manual calibration is often used in land surface modelling. Starting from the parameter set that is already used for CMIP6 (Boucher et al., 2020), we found that the simulated ET is underestimated compared to GLEAM and FLUXCOM, and Q is overestimated compared to HydroPortail over France because of different inputs, spatial resolutions, and evaluation datasets used in our study. Thus, we need to calibrate ORCHIDEE to reduce the biases by increasing simulated ET and decreasing simulated Q. This is the general philosophy of our calibration method.

Besides, it is unclear why these specific parameters were selected for testing (the model probably has many other parameters) and if the modifications apply uniformly over the entire testing zone.

Response: As we can not test the sensitivity of all parameters (more than 500 parameters to test) in ORCHIDEE to ET and Q, based on previous literature (e.g.,

Kiliç et al., 2023; Raoult et al., 2021; MacBean et al., 2020; Dantec-Nédélec et al., 2017; Campoy et al., 2013) and expert experience, we chose the sensitive parameters that regulate surface roughness, soil hydraulics, and vegetation morphology (detailed in Section 2.1) to improve the ET and Q simulation results (detailed in Section 3.2).

And yes, the modifications are applied uniformly over the entire simulation area.

It is also unclear how the criteria calculated at each station were aggregated to get an overall performance at the level of the catchment set (e.g. the KGE criterion may generate highly negative values which may bias the calculation of the mean performance), which weight was given to ET and Q respectively during calibration (i.e. how the authors cope with the multi-objective aspect of calibration), and how the criteria were actually used in the calibration process.

Response: We do not calculate the mean values of the performance criteria but the medians and quartiles via boxplots to summarize their full distribution.

Since manual calibration is used in our study (no optimization), we don't need a function to combine all the performance criteria as one objective function. Instead, we consider all the information from performance criteria and we eventually choose the parameter set with experience and judgment even though there are some compromises to make.

12. Section 2.4: Another problem in the experimental design is that the authors only report performance criteria in calibration. The authors do not test how the model would behave if only half of the available period had been taken for calibration and the other half for validation (as classically done in a split sample test scheme) or if the catchment set had been split in two parts, one for calibration and the other for spatial validation (proxy-basin test). This is essential to evaluate the robustness of the proposed modelling options.

Response: Thank you for raising the issue of modelling robustness.

To assess the robustness of our calibrated model, the paper included trend analyses (Section 3.3.3), which show that the model captures long-term changes satisfactorily over a large proportion of stations.

Following the reviewer's suggestion, we also conducted split sample tests on Q simulations over France for the 6 calibration experiments with 3 separate periods (first half of Q time series, second half of Q time series and total Q time series) as shown in figure below. These split sample tests show that the performances of Q simulations are stable over the three periods.

[Figure]

We will add a sentence in Section 2.3: "Split sample tests were also performed and showed stable performances if the Q time series are split in two halves (Section S11 in supplementary material)."

We will also add the above information in supplementary material.

13. Table 1: I did not understand how the authors selected the PFT classes whose values were modified. Why only six classes over the 15 PFT were modified?

Response: These six classes are the ones that are present in France. This will be mentioned in L225.

Why only the bias with FLUXCOM appears in the table? Does it mean that only the bias against this ET product was considered during calibration?

Response: We considered both GLEAM and FLUXCOM products during calibration to compare with the ORCHIDEE simulation, under the assumption that these two products offer a plausible range for ET. To avoid confusion, we will add the bias of simulated ET against GLEAM in Table 1.

14. Section 3.1: I did not understand where the "true" values of catchment area come from.

Response: The reference basin area values come from the HydroPortail dataset as mentioned in Figure 2. Following this comment, we will change the sentence to "Figure 2 shows the good performance of the high-resolution river routing model in

simulating the basin areas across France, with $R^2$= 0.999 across the 3507 stations compared to the information from HydroPortail."

15. L264-265: Time lags of -3 to 5 days remain very large for the French catchments. How such errors can be obtained?

Response: Many stations with a time lag greater than a few days in absolute value correspond to catchments smaller than 10000 km² (Fig. 5). These time lags might be large for flood events. However, our study focuses on the general ability of ORCHIDEE to represent reliable water budgets temporally and spatially at daily time step over France, and the trends for historical analysis and climate change impact assessments. This could be improved but it is not in the scope of this study. In addition, most catchments larger than 10000 km² (Fig. 5) do not have large time lag errors, and we have managed to decrease the time lag error from -11~27 days to -3~6 days (error divided by a factor 4 approximately) from STD to EXP4.

Such errors can be sourced from uncertainties of model structure (e.g., simple routing and groundwater module in ORCHIDEE), parameterisation (same parameter values for all the grid cells across France and uniform velocity parameters in routing scheme across France), and Safran inputs.

To clarify our results, (1) we will change L318-320 to "In terms of the time lag criterion, most simulated Q with larger time lag errors correspond to catchments smaller than $10^4$ km² (Figure 5), with simulated Q leading the observations by 2 to 6 days in the Seine River basin, but lagging the observations by 2 to 4 days in the Loire River basin." (2) we will change the sentence in L341-342: "This degrades the ORCHIDEE's performance (e.g., bias and time lag) in the sedimentary basins. This problem could be approached by assigning larger residence times to the slow reservoirs of grid cells in sedimentary basins. However, it is difficult in practice because parameters are applied uniformly over France."

Is there a problem of calculation of this criterion for catchments with slow response?

Response: Yes, we have such errors in groundwater dominated catchments (the northern part of France such as the Seine river basin and the Somme river basin) with simulated river discharge leading the observations.

How such errors can be obtained and how we can improve the simulations are explained above.

16. Section 3.3: Though results shown here seem to bring some improvement over the standard model version, one strong limit of the model evaluation shown here is that it is very difficult to say if the results are satisfactory or not. Some errors seem

still very large after improvement (e.g. time lag in some cases). The use of an external benchmark (e.g. a simpler model) would be very useful to discuss this point.

Response: A model is an approximation of reality and never perfect. We discussed the uncertainties of our simulations and pointed out the room for improvement for future studies in Section 4.

In the Explore2 project, we have simulations from 8 other hydrological models including semi-distributed rainfall-runoff models, fully distributed rainfall-runoff models, and land surface models. The report (in French) has not been published yet. It shows that simpler rainfall-runoff models can yield better Q simulations in terms of bias and KGE, but trend accuracy was not tested, nor the reliability of ET simulation.

17. Section 3.3: Maybe the title of this section should be changed to "Spatial evaluation…" since it presents this part of the evaluation. Section 3.2 and 3.3 are basically based on the same results obtained in calibration. The titles should not suggest that one part is calibration and the other is an independent evaluation, to avoid confusion.

Response: Thank you for pointing out the confusion of the section titles. We hope the following ones will be clearer:

"3.2 Calibration results" to "3.2 Performance of the different experiments";

"3.3 Evaluation of simulated water fluxes" to "3.3 Preferred experiment: Spatial evaluation of the simulated water fluxes";

"3.3.3 Q trend performance" to "3.4 Preferred experiment: Evaluation of river discharge trends"

18. Section 3.3.1: Figures 4 a and b suggest that actual evapotranspiration is very difficult to know. The maps show huge differences in some regions (and I think the sentence in L297-298 is wrong).

Response: Figures 4 a-b show the mean ET estimates from GLEAM and FLUXCOM products. It is true that there are large differences of ET estimates by these two products because they are based on different methods (see L198-202).

To avoid confusion, we will change the sentence in L297-298 to "Both the GLEAM and FLUXCOM datasets are used to evaluate the ET simulation by ORCHIDEE in this study. Both datasets show more ET in the southern part (except for the high Alps) and less ET in the northern part of the simulation domain (Figure 4a-b)."

In these conditions, I do not understand how these products can be used simultaneously to constrain the calibration, or at least how the choice can be made between the two maps to select the best model parameters.

Response: We can consider, however, that ORCHIDEE, GLEAM and FLUXCOM provide independent estimates of ET, all with their own uncertainties (Liu et al., 2023). In this framework, our guideline was to compare the ORCHIDEE simulation with these products, which offers plausible range.

Eventually, in our study, we need to compromise between these products to define an acceptable parameter set in our manual calibration procedure given the inherent uncertainties.

Following this comment, we will add a sentence in L203: "GLEAM and FLUXCOM provide independent ET estimates, both of them with large uncertainties (Liu et al., 2023). They are used in combination to approach the plausible range of observed ET."

19. Section 3.3.1: Maybe I am wrong, but some regions where there is a large bias seem to correspond to zones where there is a lower density of stations (Fig. 2). It this something observed by the authors? If yes, this may suggest that there are problems in transposing the parameter sets in space.

Response: The two ET products are gridded products, and they are independent from the stations.

20. L307-309: Does the larger average bias with FLUXCOM comes from the fact that it was not directly considered in the calibration? I don't know whether the more consistent spatial bias with FLUXCOM is good news. Please comment on this.

Response: We manually calibrated ORCHIDEE step by step to reduce ET biases considering both GLEAM and FLUXCOM to define a plausible range.

Both GLEAM and FLUXCOM products are widely used in literature to calibrate models. It is good news that the simulated ET is spatially consistent with FLUXCOM, but we cannot say why it is more consistent with FLUXCOM than with GLEAM since ET from ORCHIDEE, FLUXCOM and GLEAM are based on different methods. And it is also good news that the bias of the simulated ET to GLEAM over the entire study domain is better than that to FLUXCOM.

Anyway, we do not give preference to any specific product because they are both helpful to calibrate ORCHIDEE in the light of ET uncertainty.

21. Section 3.3.2: I was surprised that the human influences appear to be one of the main reasons mentioned for model failure. Though they probably contribute to the limited performance sometimes, I doubt that the level of influences on these basins can explain the gaps between observed and simulated time series. This is not realistic.

Response: We showed that human influences degrade the goodness-of-fit indicators especially for correlation and KGE of Q (Appendix I). Besides, many publications have demonstrated that human influences deeply changed the river flow dynamics in France (e.g., François et al., 2014; Flipo et al., 2020).

Besides there are many stations in the catchment sample that are influenced. Why were they kept as calibration target if the objective is to simulate natural behaviour? The calibration process and catchment selection should be better explained and potentially revised.

Response: Following this suggestion, we will (1) add a sentence in L112 "It must be noted that this version of ORCHIDEE does not include any human impact, with the exception of the presence of crops among the possible vegetation types." (2) add more information in L206 of Section 2.3: "Although the version of ORCHIDEE used in this study does not include any human impact, the 1785 selected hydrometric stations were all used in the evaluation process, whether human influenced or not. This enables a more comprehensive assessment of ORCHIDEE, as natural or weakly influenced stations are few in number (only 536, see Section S9 in supplementary material) and exclude the stations along the main streams of the four major French rivers (Seine, Loire, Garonne, and Rhône)."

Besides, we have also pointed out the inclusion of human activities (e.g., dams and irrigation) in the ORCHIDEE LSM for future work in L434-438 of Section 4.

22. L319-320: 2 to 4 or 6 days timelag is huge for these basins. In practice, how can the model be used with such time lags?

Response: See the comment N°15.

23. L361-369: This probably would be better placed in the method section.

Response: Yes, we will move to Section 2.3 in the revised version.

24. L372-383: I do not understand why human influences are no more a problem here to evaluate trends though they were one of the major reasons for model failure a few paragraphs above. For me this is not really consistent.

Response: We have the analysis of human impact on trends in L349-360.

25. Section 4: As explained in previous comments above, I think that the discussion should better acknowledge the limitations of the modelling framework proposed here. Though it was improved, the model is still limited in some cases (as any model).

Response: Thank you for this suggestion. We analyzed the limitations of our work in this section and pointed out the room for improvement for future work. Maybe the original version (L406-438) is not clear and to make it more readable, L406-438 of Section 4 will be changed to:

[revised manuscript text omitted]

Besides if the authors intend to do an actual model calibration, they should do corresponding validation test to evaluate model robustness in space and time. Else the results probably show an over-optimistic picture of model predictive power.

Response: We did not use an optimization based calibration method. Besides, we conducted split sample tests for your information (see comment N°12).

26. L384: I do not know what "high-resolution" means here. There are models implemented at the km² scale.

Response: We will change "high-resolution" to "high-resolution of approximately 1.3 km".

27. Appendices: There are a lot of appendices. I am unsure they should be kept as appendices. They may be better placed in supplementary material.

Response: We agree and we will move these appendices to supplementary material.

28. Appendix I: Appendix I is not an actual demonstration that the modelling problems come from the artificial influences. Some other characteristics which may differ between the two sub-samples may also explain the performance differences.

Response: We were not trying to attribute all modelling problems to human influences. Appendix I is an example to show that ORCHIDEE performs better for natural or weakly influenced catchments. We have also discussed other reasons like model structure, parameterization and input data in Section 4.

Minor comments
29. Introduction: a few subtitles may be useful to highlight the main aspects of the introduction

Response: Thanks for the suggestion. We will add subtitles "Land surface models for high-resolution hydrological simulations" for paragraph 1-3, "How to calibrate land surface models?" for paragraph 4-5, "How to evaluate the performance of land surface models?" for paragraph 6, and "Aim and novelty of the study" for paragraph 7-8.

30. L102: Please clarify what is "revision 7738".

Response: Thanks for the suggestion. This will be deleted from the title, but for clear documentation of the code (refer to EGU guidelines), we will change L103-105 to "The ORCHIDEE model is a physically-based LSM developed at the Institut Pierre Simon Laplace (IPSL) as the land component of the IPSL climate model, which is used for all the past and future climate simulation exercises carried out for the IPCC reports as part of the Coupled Model Intercomparison Project (CMIP) (IPCC,2023). Here, we use ORCHIDEE version 2.2 (with revision 7738), which is very close to the version used as the land component of the IPSL-CM6 climate model (Boucher et al., 2020; Cheruy et al., 2020)."

31. Fig. 1: I wonder whether this figure is actually useful (at least in the main text)

Response: We think Fig 1 is useful in the main text to have a general idea of French geography and hydrography for non-French people.

32. Fig. 3: The caption should indicate which distribution percentiles are shown on the box-plots.

Response: We will add the sentence "For each boxplot, the lower and upper hinges are the first and third quartiles; the minimum and maximum values extend from the first/thrid hinge to 1.5 times of the inter-quartile range. "

**Final note: We will also correct the style of some sentences so that they are more readable, and orthographic and grammar errors.**